# Transient neuronal suppression for exploitation of new sensory evidence

Maxwell Shinn [1,2], Daeyeol Lee [3,4,5,6], John D. Murray [1,2,7,8✉] & Hyojung Seo [1,2,8✉]

In noisy but stationary environments, decisions should be based on the temporal integration of sequentially sampled evidence. This strategy has been supported by many behavioral studies and is qualitatively consistent with neural activity in multiple brain areas. By contrast, decision-making in the face of non-stationary sensory evidence remains poorly understood. Here, we trained monkeys to identify and respond via saccade to the dominant color of a dynamically refreshed bicolor patch that becomes informative after a variable delay. Animals' behavioral responses were briefly suppressed after evidence changes, and many neurons in the frontal eye field displayed a corresponding dip in activity at this time, similar to that frequently observed after stimulus onset but sensitive to stimulus strength. Generalized drift-diffusion models revealed consistency of behavior and neural activity with brief suppression of motor output, but not with pausing or resetting of evidence accumulation. These results suggest that momentary arrest of motor preparation is important for dynamic perceptual decision making.

[1] Interdepartmental Neuroscience Program, Yale University, New Haven, CT 06520, USA. [2] Department of Psychiatry, Yale University, New Haven, CT 06520, USA. [3] Zanvyl Krieger Mind/Brain Institute, Johns Hopkins University, Baltimore, MD 21218, USA. [4] Kavli Discovery Neuroscience Institute, Johns Hopkins University, Baltimore, MD 21218, USA. [5] Department of Psychological and Brain Sciences, Johns Hopkins University, Baltimore, MD 21218, USA. [6] Department of Neuroscience, Johns Hopkins University, Baltimore, MD 21218, USA. [7] Department of Physics, Yale University, New Haven, CT 06520, USA. [8] Department of Neuroscience, Yale University, New Haven, CT 06520, USA. ✉email: john.murray@yale.edu; hyojung.seo@yale.edu

Momentary evidence from sensory stimuli or memory seldom provides sufficient information to choose the most appropriate action. Rather, speed and accuracy of choice behaviors in humans and animals are more consistent with models of integrators or accumulators, such as the drift-diffusion model (DDM), in which noisy evidence is temporally integrated as a decision variable that triggers an action upon crossing a threshold[1,2]. In addition, neural activity in multiple brain areas might correspond to the trajectory of such decision variables[3,4]. Yet, how this relatively simple strategy can be extended for time-varying sensory stimuli[5] remains poorly understood. While several studies have utilized time-varying stimuli[5–10], it is unknown how moment to moment changes in sensory inputs impact evidence integration. In the present study, we investigated how the trajectory of neural activity related to evidence accumulation might be adjusted by the subtle onset of sensory signals decoupled from the onset of the stimulus itself.

We examined three different hypotheses regarding cognitive strategies for how the detection of informative stimuli might impact perceptual decision making. First, the neural activity related to the decision variable, such as that observed in lateral intraparietal cortex (LIP) and frontal eye field (FEF), shows a temporary dip after stimulus onset[11–16], and this has been interpreted as the reset of an integrator for the decision variable. According to this "reset" model, the decision variable and its neural correlate might be fully or partially reset when the non-informative sensory stimulus is replaced by an informative stimulus. Second, rather than a reset, the decision variable might be temporally frozen so that the incoming stream of evidence during a volatile transition period can be ignored. Indeed, such a "pause" model has successfully accounted for the reaction time (RT) data during behavioral tasks, such as countermanding or double-step saccade tasks, in which the subjects must adapt to sudden and unpredictable changes in instructions[17–21]. Finally, decision makers might adapt to the unpredicted arrival of new information simply by suppressing their motor outputs[22–24] without modifying the state of decision variables, similar to models proposed in other systems in humans[17,25–27]. Unlike the reset or pause models, this "motor suppression" model predicts that even motor outputs unrelated to the task, such as microsaccades, might be suppressed.

We tested these three alternative hypotheses by analyzing the behavioral data and neural activity recorded from the FEF in monkeys performing a perceptual decision-making task in which the stimulus onset was temporally decoupled from the onset of informative stimulus evidence. After the onset of the informative stimulus, we observed transient suppression in saccadic motor output, as well as in FEF activity at a population and single-neuron level. Moreover, suppression of motor output and FEF activity was greater for stronger sensory signals, resulting in a negative relationship of the RT distribution and FEF activity on coherence. This is in contrast to the usual positive relationship of coherence with these measurements. Formal model testing showed that both behavioral and neural data were most consistent with the motor suppression model compared to the reset and pause models. These results suggest behavioral and neural signatures of motor suppression as a cognitive mechanism for the strategic use of changes in evidence during perceptual decision-making.

## Results

**Immediate effect of a change in evidence on the RT distribution**. We trained two rhesus monkeys to perform a two-alternative forced-choice color-discrimination task (Fig. 1). The stimulus for discrimination was a square patch consisting of blue and green pixels, and the relative number of pixels between the two colors, referred to as color coherence, determined the difficulty of discrimination. To temporally dissociate the processes associated with the detection of a change in the evidence from those for the detection of the onset of the stimulus itself, stimulus presentation was divided into two consecutive periods containing an uninformative "presample" and informative "sample".

During the variable presample period (0, 400, or 800 ms), there were equal numbers of blue and green pixels displayed in the stimulus, corresponding to a color coherence of zero. During the sample period, color coherence of the stimulus changed to a non-zero value, which was fixed for a single trial but randomly selected from three values across trials. For comparison, we also included a "zero-coherence" condition, in which an equal ratio of blue and green pixels was maintained throughout the entire trial. No explicit cue was presented to indicate the abrupt transition from presample to sample, and pixels were rearranged at 20 Hz during both periods to make this transition non-obvious. Reward cues surrounded the saccade targets and indicated whether a correct response to the designated target would result in a large or small reward (see Methods). Only correct choices made after the sample onset were rewarded. The animals' performance in this task changed with color coherence and presample duration. As reported in detail previously[28], choices were less accurate (Fig. 1c, Supplementary Fig. 1a) and slower (Fig. 1d, Supplementary Fig. 1b) during trials with a low coherence.

To gain a mechanistic understanding of how the monkeys might be performing the task, we fit a generalized drift-diffusion model (GDDM)[29] to RT distribution data, a model previously shown to explain both the reward bias and timing in our dataset[28]. This model incorporated leaky integration, an urgency signal, and two forms of reward bias. As reported previously[28], the GDDM predicts higher accuracy and shorter RT in trials with higher color coherence, and this was confirmed in the data (Fig. 1c, d; Supplementary Fig. 1a, b).

Despite the general success of the GDDM in accounting for the complex behavioral patterns observed in the RT and choice data, we found a behavioral feature immediately after an abrupt change in evidence which cannot be explained by the model. To examine the effect of changing evidence on the most rapid responses, we focused on the RT during trials with the longest presample duration, because trials with shorter presample durations contained few responses within a 200-ms window after sample onset (Fig. 2a, c), presumably related to fewer responses driven by a slowly ramping urgency signal[28–30]. We found that, despite an increase in evidence, a change to a higher coherence resulted in a short-latency suppression of responses, visible as a "dip" in the RT distribution (Fig. 2b, d). This RT dip was more pronounced for larger changes in evidence, and its latency was similar to the results reported previously[23,31,32]. The dip was present for a wide range of bin sizes (Supplementary Fig. 2). The coherence-dependence also implies it is not the result of anticipating the change. This dip in the RT distribution is inconsistent with the standard models of evidence accumulation such as the DDM, as well as the GDDM, which all predict that stronger evidence will shorten RTs without any dip in the RT distribution (Fig. 2e).

**Coherence-dependent dip in FEF population activity**. Activity in cortical areas such as FEF and LIP is often hypothesized to represent the accumulation of relative evidence favoring a particular behavioral response. According to this hypothesis, an increase in evidence favoring the target in the response field of a neuron should lead to an increase in the neuron's firing rate. As in the analysis of RT data, we focused on FEF activity during the

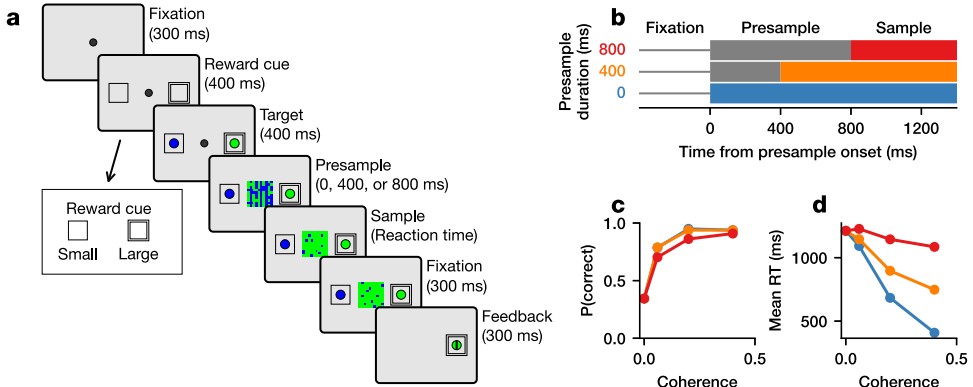

**Fig. 1 The color-discrimination task. a** The temporal sequence of trial events in the color-discrimination task. Inset in the lower left are cues which indicate a large or small reward. **b** Schematics for the time course of color coherence throughout the trial for each presample duration. Gray indicates zero coherence, and colors indicate non-zero coherence. The thin gray horizontal lines denote the fixation period. **c** Psychometric function showing the mean probability of a correct response for Monkey 1. Error bars representing standard error of the mean are hidden beneath the markers and are based on 28,378 trials. **d** Chronometric function showing the mean RT as the function of coherence for Monkey 1. Error bars representing standard error of the mean are hidden beneath the markers and are based on 28,378 trials.

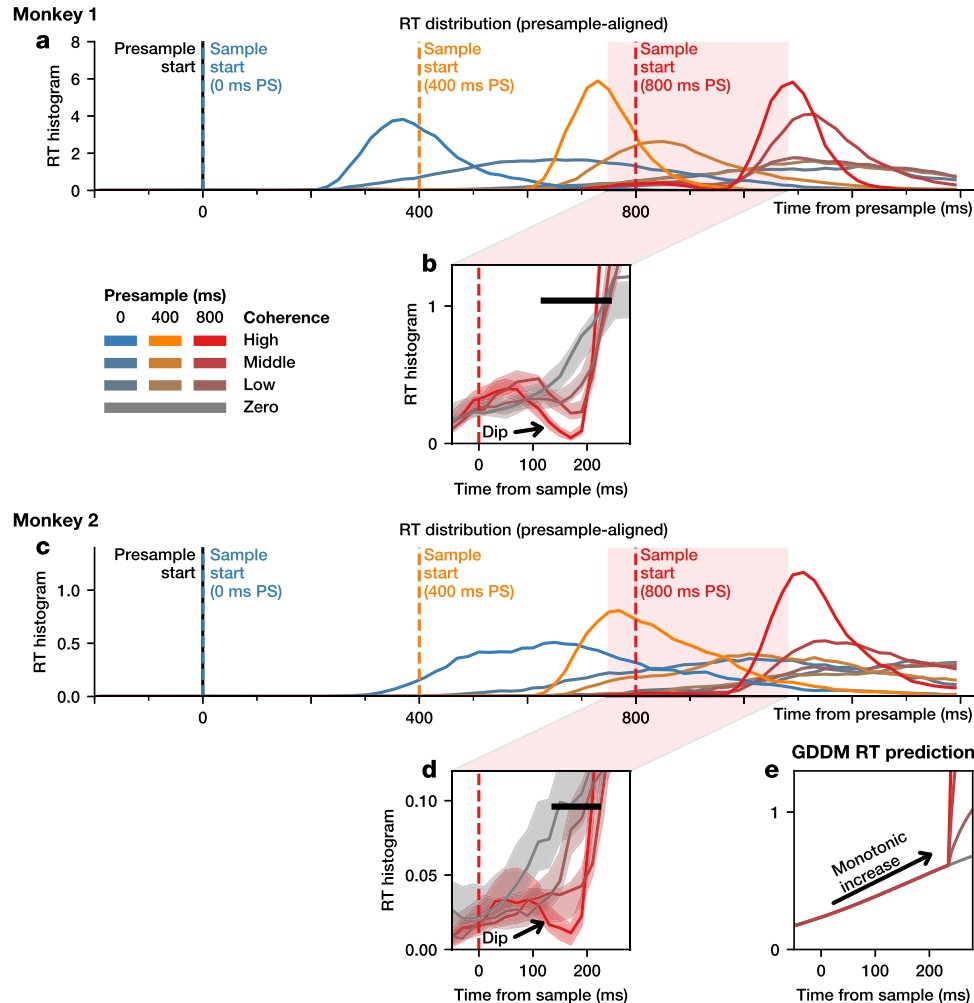

**Fig. 2 Transient effect of changes in evidence on RT.** The RT distribution for all trials, aligned to the presample onset, for Monkey 1 (**a**) and Monkey 2 (**c**). The RT distribution centered around the onset of the sample during trials with 800-ms presample for Monkey 1 (**b**) and Monkey 2 (**d**). **e** Predictions from a generalized drift-diffusion model (GDDM) for the portion of the RT distribution shown in (**b, d**). The black bar indicates significance ($p < 0.05$, one-tailed test for a decrease in the mean, bootstrapping across trials). Shaded regions represent bootstrapped 95% confidence interval of the mean. RT distributions are smoothed for visualization only, with significance tests performed before smoothing.

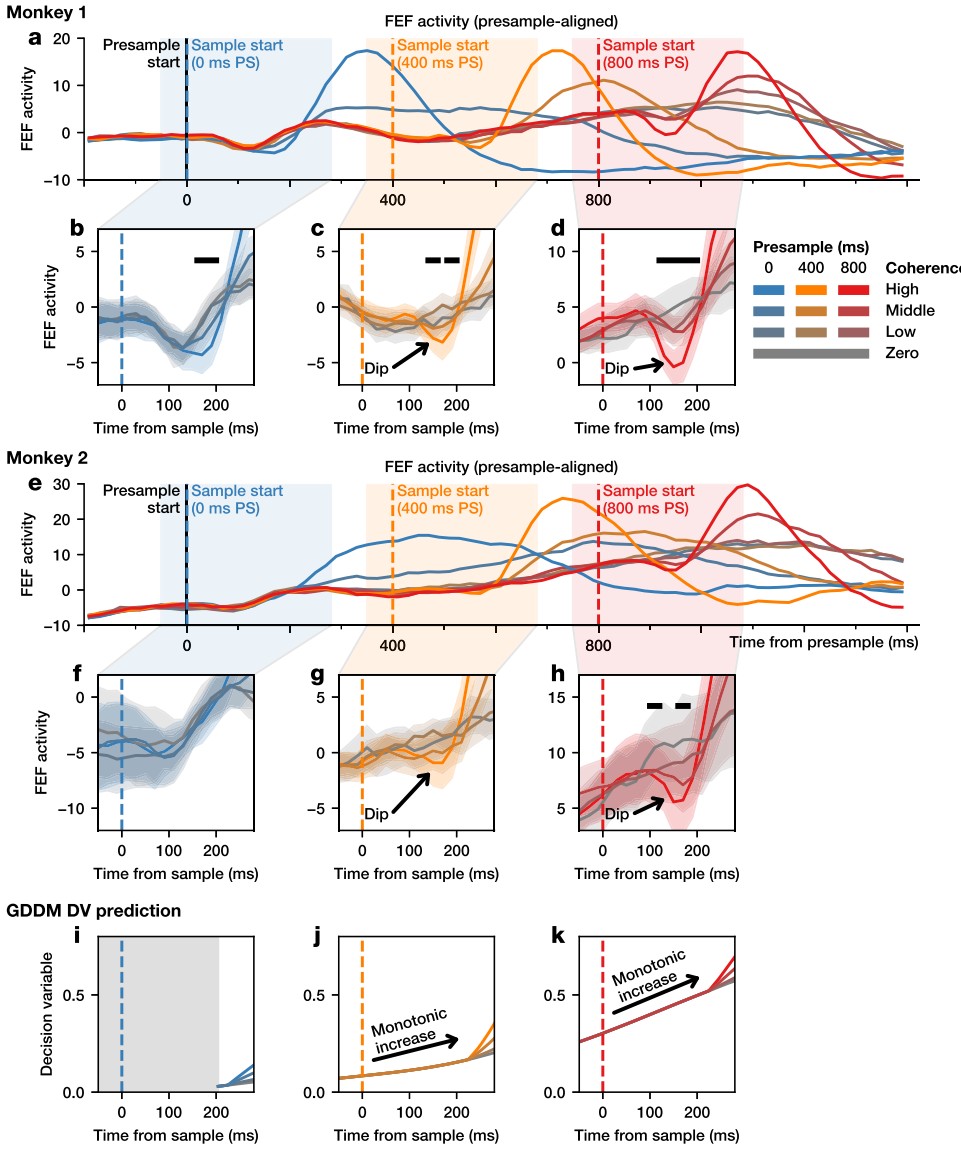

**Fig. 3 Transient effect of changes in evidence on population FEF activity.** The normalized population activity for all trials, aligned to the presample onset, for Monkey 1 (**a**) and Monkey 2 (**e**). **b–d**, **f–h** Highlighted is activity centered around the onset of the sample during trials with a 0- (**b**, **f**), 400- (**c**, **g**), and 800-ms (**d**, **h**) presample duration for Monkey 1 (**b–d**) and Monkey 2 (**f–h**). The black bar indicates significance ($p < 0.05$, one-tailed test for a decrease in the mean, bootstrapping across neurons). Shaded regions represent bootstrapped 95% confidence interval of the mean. Predictions from a drift-diffusion model decision variable (DV) are shown below for the 0- (**i**), 400- (**j**) and 800-ms (**k**) presample durations. Light gray indicates no prediction within the model's non-decision time. Activity is smoothed for visualization only, with significance testing performed before smoothing.

period immediately following sample onset, and examined mean-normalized activity averaged across all FEF neurons separately for each presample duration and coherence level.

During the trials with 400- or 800-ms presample duration, we found that FEF neurons showed a robust coherence-dependent reduction in their activity, such that higher coherence resulted in larger reduction in activity (Fig. 3c, d, g, h). The latency of this "evidence dip" in FEF activity was comparable to that of the dip in the RT distribution. Consistent with previous findings, a dip in FEF activity was also seen immediately after the onset of the stimulus itself in trials without a presample period (0-ms presample duration). In comparison to the evidence dip, we refer to this as the "stimulus dip", because unlike the evidence dip observed after the presample, the magnitude of the stimulus dip was largely independent of the color coherence of the stimulus (Fig. 3b, f). Similar to the behavioral results, the presence of the evidence dip is not predicted by models of evidence

accumulation. If FEF activity exclusively represents the total accumulated evidence, then it should not decrease when stronger evidence becomes available. Likewise, the mean trajectory of the decision variable simulated with the GDDM did not show a reduction in activity at any point in the first 300 ms after sample onset (Fig. 3i-k).

**Coherence-dependent dip in individual FEF neurons.** We next examined whether the evidence dip could be detected at the single-neuron level. We first used a linear regression model to determine how single-neuron activity was modulated by the experimental variables such as evidence (color coherence), reward, and choice, at the onsets of the presample, sample, and saccade (Eq. (1)). We found that reward magnitude significantly modulated the activity of almost all neurons at all three time points, and that the animal's choice significantly modulated neural activity at the time of saccade onset (Supplementary

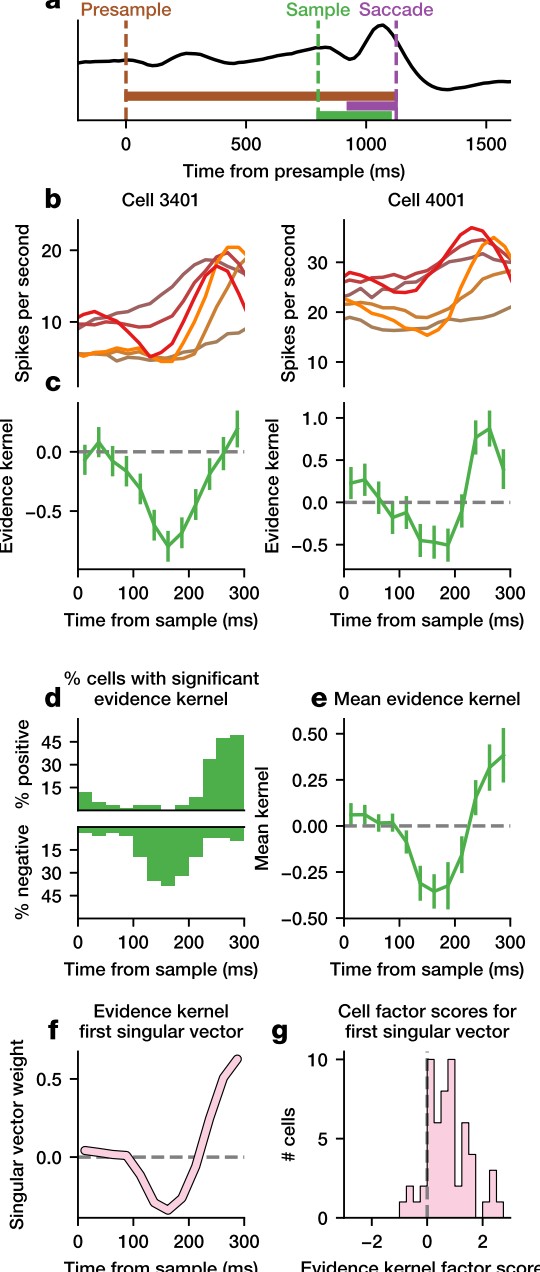

**Fig. 4 Evidence dip in individual FEF neurons for Monkey 1. a** A schematic of the regression model showing alignment of the kernels to the presample, sample, and saccade. **b** Smoothed firing rate for two example neurons across conditions. Colors are the same as in Figs. 2–3. **c** Evidence kernel for the example neurons in (**b**). Error bars indicate 95% confidence interval of the regression coefficient. **d** For each point in time, the number of neurons in the population with significantly positive (top) or negative (bottom) evidence kernel is shown ($p < 0.05$, one-tailed $t$ test). **e** Mean evidence kernel across neurons. Error bars indicate 95% confidence interval of the mean. **f** The first singular vector of the evidence kernels is shown, along with (**g**) the corresponding factor scores of each neuron. Analyses are based on 57 neurons.

Fig. 3). In addition, at the sample onset, but not at the presample or saccade onset, neural activity was significantly modulated by coherence, such that the mean firing rate decreased with coherence, consistent with the dip we described above.

To explore whether this early modulation by coherence after the sample onset took the form of a dip, we fit a time-resolved regression model to the instantaneous firing rate of each neuron (Eq. (2)). Our model included multiple kernels aligned to the sample onset, presample onset, and saccade onset. Consistent with the results from the analysis of the mean activity (Supplementary Fig. 3), we made the presample-aligned kernel sensitive to reward magnitude, the sample-aligned kernel to coherence, and the saccade-aligned kernel to choice (Fig. 4a). Here, we focus on the sample-aligned kernel scaled by coherence (referred to as evidence kernel) and the presample-aligned kernel (referred to as stimulus kernel). Thus, the evidence kernel represents the time course of coherence-dependent neural activity after the transition from presample to sample, and the stimulus kernel represents mean neural activity after stimulus onset, corresponding to the evidence dip and stimulus dip respectively, as described for the population activity. The evidence kernel allowed us to examine the effect of coherence immediately after the onset of the sample independently of saccadic activity.

During the sample period, neural activity reflecting the value of the decision variable should increase with coherence, namely, the evidence kernel should always be non-negative. Contrary to this prediction, we found that for many FEF neurons, activity immediately after sample onset tended to decrease more for larger increases in the coherence of the sample stimulus favoring the action towards the neuron's response field, resulting in a negative evidence kernel. Some neurons show dip-like activity traces (Fig. 4b, Supplementary Fig. 4b), and their evidence kernels were negative between 100 and 200 ms after the sample onset (Fig. 4c, Supplementary Fig. 4c). Across the population, many more neurons show significant negative evidence kernel during the same period. For example, in the interval 150–175 ms after the sample onset, 39% vs. 0% of neurons for Monkey 1, and 52% vs. 9% of neurons for Monkey 2, show significant negative vs. positive evidence kernel, respectively ($p < 0.05$, one-tailed $t$ test; Fig. 4d, Supplementary Fig. 4d). The fraction of neurons with significantly positive evidence kernel eventually began to increase about 200 ms prior to the saccade onset. Consistently, the average evidence kernel across all neurons also shows a significant dip in this time interval (Fig. 4e, Supplementary Fig. 4e).

To confirm that the mean evidence kernel reflects the temporal pattern of a dip across single neurons, we computed the best rank-one approximation of the kernels, the first singular vector (Fig. 4f, Supplementary Fig. 4f). We found that the temporal profile of weights for the first singular vector is highly correlated with the mean evidence kernel (Monkey 1 $r = 0.970$, Monkey 2 $r = 0.998$). This singular vector has a predominantly positive factor scores across neurons (Monkey 1, 89% neurons, Monkey 2, 100% neurons; Fig. 4g, Supplementary Fig. 4g), indicating that the dip exhibited by the mean kernel is representative of the kernels of individual neurons. The presence of the dip in the evidence kernel suggests that the evidence dip is coherence-dependent. To confirm this is the case, we performed the above analysis on the coherence-independent sample-aligned kernel and found no evidence of a dip (Supplementary Fig. 5).

We also tested whether the evidence dip in FEF activity might be confounded by saccade-related activity, using three different types of analyses. First, we extended the time-resolved regression model to increase the lengths of the saccade and evidence kernels and make the saccade kernel sensitive to coherence (Eq. 3; Supplementary Fig. 6a). We found that the evidence dip was still present in this extended regression analysis (Supplementary Fig. 6b, h). This extended model revealed a broad dip of different form and timing than the evidence dip in its coherence-dependent saccade kernel (Supplementary Fig. 6c, i), which is

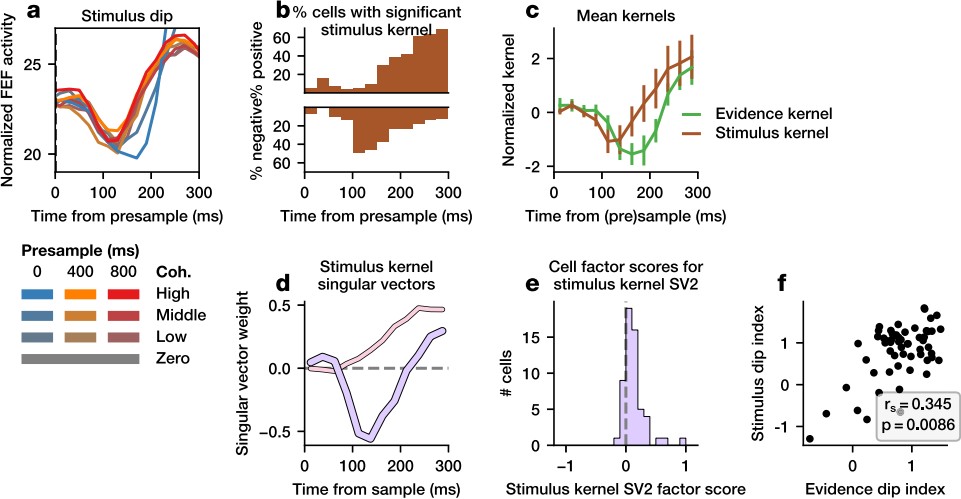

**Fig. 5 Comparison of evidence and stimulus dips for Monkey 1. a** Mean FEF activity is plotted for each presample and coherence condition at presample (400 or 800 ms presample) or sample (0 ms presample) onset. **b** The fraction of neurons at each point in time with significantly positive (top) or negative (bottom) stimulus kernel ($p < 0.05$, one-tailed $t$ test). **c** Mean stimulus kernel is shown with overlaid evidence kernel. Time is given as time since the sample (evidence kernel) or presample (stimulus kernel). Error bars indicate 95% confidence interval of the mean. **d** The first (SV1) and second (SV2) singular vector of the stimulus kernels are shown, along with (**e**) the corresponding factor scores for each neuron on SV2. **f** Evidence dip index is plotted against stimulus dip index. Spearman correlation and two-tailed p-value is inset. Analyses are based on 57 neurons.

expected from a broader RT distribution and FEF activity that builds up more slowly during trials with low coherence.

Second, we also tested how the timing of the evidence dip was related to RT. We divided trials into RT quintiles, and determined the timing of the evidence dip for each quintile (see Methods). If the evidence dip were saccade-related, there would be a positive correlation between evidence dip latency and RT quintile. We found that the timing of the evidence dip did not vary significantly with RT (Kendall tau = −0.10 for Monkey 1 and 0.02 for Monkey 2, $p > 0.05$; Supplementary Fig. 6d, f, j, l). We also tested whether the saccade-aligned kernel might show a dip, and examined how the timing of this saccade-related dip might be related to RT. For Monkey 2, the saccade-related kernel did not show a robust dip (Supplementary Fig. 6k). For Monkey 1, we reliably detected a dip in the saccade kernel, and the latency of this dip showed a negative correlation with RT (Kendall tau = −0.79, $p < 0.001$; Supplementary Fig. 6e, g), suggesting that its timing was largely locked at the sample onset.

Third, we examined whether the pattern of neural activity associated with the dip is correlated across neurons with the pattern associated with other task elements, such as motor response or reward bias. We computed the Spearman correlation between the non-time-resolved regression coefficients from Supplementary Fig. 4. If these patterns covary across FEF neurons, we would expect a significant positive or negative correlation between the coefficient for coherence at sample onset and either the coefficient for choice at saccade onset or reward at presample onset. However, there was no significant correlation with motor (Spearman r = −0.11, $p = 0.34$) or reward (Spearman r = −0.20, $p = 0.07$). These results demonstrate that the evidence dip is unlikely to reflect pre-saccadic activity.

**Coherence-dependent and -independent dips are correlated.** Previous studies have reported a dip in neural activity in LIP and FEF when the stimulus, such as randomly-moving dots, is first presented. Indeed, we found a similar dip in FEF activity immediately after the stimulus was first presented regardless of whether it was presample or sample, e.g., in the 0-ms presample

condition (Figs. 3b, f; 5a, Supplementary Fig. 7a). We examined whether this stimulus dip might show the same properties, in the same neurons, as the coherence-dependent evidence dip. First, we found that ~100–200 ms after presample onset, many FEF neurons exhibited a significantly negative stimulus kernel but very few neurons exhibited positive stimulus kernel ($p < 0.05$, one-tailed $t$ test; Fig. 5b, Supplementary Fig. 7b). For example, during the interval 100–125 ms after stimulus onset, 49% vs 5% of neurons for Monkey 1, and 35% vs 17% of neurons for Monkey 2, showed a significant negative vs positive kernel. There was also a dip in the mean stimulus kernel (Fig. 5c, Supplementary Fig. 7c, brown). To determine whether individual neurons exhibited a similar dip, we computed the first two singular vectors of the stimulus kernels. While the first singular vector showed a monotonic ramp (Fig. 5d, Supplementary Fig. 7d, pink), the second singular vector resembled a dip (Fig. 5d, Supplementary Fig. 7d, purple) which had high correlation with the mean stimulus kernel (Monkey 1 r = 0.856, Monkey 2 r = 0.499) and predominantly positive factor scores on individual neurons (Monkey 1, 82%, Monkey 2, 83%; Fig. 5e, Supplementary Fig. 7e), demonstrating that individual neurons showed a stimulus dip.

Next, we compared the stimulus dip to the evidence dip. The timing of the stimulus dip was slightly earlier than the evidence dip (Fig. 5c, Supplementary Fig. 7c), with a significantly earlier minimum of the mean stimulus kernel than the mean evidence kernel (50 ms, $p < 0.0001$ for Monkey 1, 100 ms, $p < 0.0001$ for Monkey 2, bootstrapped two-tailed confidence interval across neurons). We then compared the magnitudes of these two dips by establishing a standardized index to quantify the dip magnitude for individual neurons. The stimulus (evidence) dip index was computed by z-scoring each stimulus (evidence) kernel, and then finding the mean value at the time points in the interval 100–150 ms (125–175 ms) (see "Methods"). As expected, this index showed a strong correlation with the factor scores on the dip-like singular vectors (Supplementary Fig. 8). We found that the evidence and stimulus dip indices were significantly correlated across FEF neurons ($p < 0.01$ in both monkeys) (Fig. 5e, Supplementary Fig. 7e), demonstrating that FEF neurons with a strong evidence dip were likely to show a strong stimulus dip.

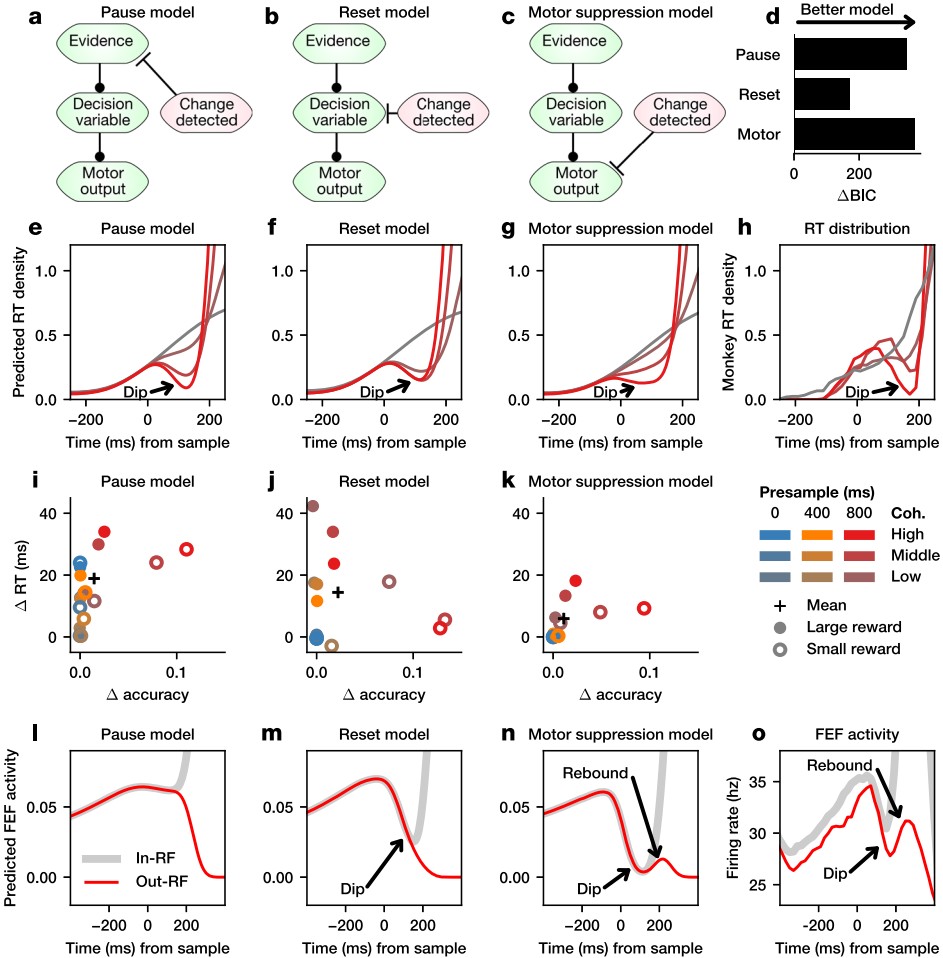

**Fig. 6 Comparison of computational models of the dip.** Schematic of the (**a**) pause model, (**b**) reset model, and (**c**) motor suppression model across stages of the decision-making process. **d** The fit of each GDDM to the RT distribution, as quantified by BIC, is shown for each of the three models. The simulated RT distribution for 800 ms presample at the time of evidence onset is shown for the (**e**) pause, (**f**) reset, and (**g**) motor suppression models. **h** The RT distribution for Monkey 1 in 800 ms presample trials at the time of evidence onset. For each coherence, presample, and reward condition, the difference in mean RT and accuracy, with and without the dip mechanism, are plotted for the (**i**) pause, (**j**) reset, and (**k**) motor suppression models. Neural predictions of the (**l**) pause model, (**m**) reset model, and (**n**) motor suppression model, based on the mean decision variable or motor-decision variable (see Methods) for inside (in-RF, black) and outside (out-RF, red) the response field on correct high-coherence trials for models fit to the RT distribution of Monkey 1. **o** Population activity from FEF neurons in Monkey 1 for correct responses inside (gray) and outside (red) the response field for high-coherence, 800-ms presample trials.

However, we found no consistent association of evidence or stimulus dip index with other neuronal properties, including mean firing rate, directional selectivity index, visuomovement index, or spike width (Supplementary Fig. 8).

**Evidence dip and motor suppression.** In order to understand the relationship between the evidence dip observed in FEF activity and the RT dip, we examined three potential mechanisms through which changes in evidence might influence behavior during perceptual decision-making. For all three mechanisms, it is assumed that a change can be detected quickly, and that the detection event triggers a downstream change in the relatively slower evidence integration pathway. We also assume that the ability to detect a change in evidence is probabilistic, with a probability of detection increasing linearly with coherence. First, in the "pause model" (Fig. 6a), when a change is detected, the stream of evidence is briefly interrupted, or "paused". This mechanism can be thought of as blocking volatile information in order to focus on a more stable evidence signal. Second, in the "reset model" (Fig. 6b), the change elicits a partial "reset" of the

decision variable back towards its initial value. This mechanism effectively discards any noise which may have been integrated during the presample period. Finally, in the "motor suppression model" (Fig. 6c), motor output is temporarily blocked, thereby suppressing responses during this period without impacting the decision variable. This mechanism ensures slow cognitive processes have sufficient time to integrate new information before making a choice. This strategy is effective because it reduces the fraction of responses occurring during the first 200 ms, which are close to chance (Supplementary Fig. 1c). For each strategy, we extended the generalized drift-diffusion model (GDDM) described previously and fit parameters to RT distributions (see Methods).

First, we show the motor suppression model provided the best fit to the RT distribution. Each of the models considered was able to fit the coherence-dependent behavioral dip in the RT distribution (Fig. 6e-h, Supplementary Fig. 9e-h). Therefore, we evaluated the fit of each model to the data using Bayesian information criterion (BIC). For comparison, we evaluated the BIC of the model without any dip mechanism present. The

improvement in the fit of the model to the monkey's behavior can be quantified by the difference in BIC between the model with and without the dip mechanism (ΔBIC). We found that the improvement in fit is greatest for the motor suppression model (Fig. 6d, Supplementary Fig. 9d). In addition, the motor suppression model is able to capture the psychometric and chronometric functions (Supplementary Fig. 1a,b). Therefore, the motor suppression model is best able to explain the dip present in the RT distribution.

Next, we show that the motor suppression model exhibits a speed-accuracy tradeoff for all conditions in the task. All three models showed a speed-accuracy tradeoff when considering mean performance in the task as a whole, resulting in increased mean accuracy but slower mean RT (Fig. 6i-k, Supplementary Fig. 9i-k, black cross). However, a general mechanism for decision-making should ideally not slow RT in environments where it does not improve accuracy. We examined this by considering each task condition separately. All three models showed the strongest improvement in accuracy for 800 ms presample trials, for which the animal is making the most random guesses, as indicated by the elevated responses in the zero-coherence condition (Fig. 2a, c). In addition, the motor suppression model resulted in slower RT only for conditions where it also improved accuracy (Fig. 6k, Supplementary Fig. 9k). By contrast, the pause and reset models slowed RT in some conditions without improving their accuracy (Fig. 6i, j, Supplementary Fig. 9i, j). This shows how the motor suppression model can serve as an efficient general mechanism for increasing accuracy without creating a generalized deficit in speed.

A comparison of the FEF population activity with the FEF activity predicted by each model also suggests that neural activity might be most consistent with the motor suppression model. For this analysis, we focused on the trials expected to show the strongest dip and the highest probability of detecting the change in evidence, namely, those with 800-ms presample duration, high coherence, and the large reward target in the neuron's response field (Supplementary Fig. 10). The pause model assumes that incoming evidence is briefly disregarded, and that the decision variable should remain fixed at its present value. Thus, it predicts a flattening of the decision variable trace and hence a flattening of FEF activity, followed by an increase for trials with the chosen target inside the response field and a decrease for those with the chosen target outside the response field (Fig. 6l, Supplementary Fig. 9l). This is inconsistent with the evidence dip in FEF. By contrast, the reset model predicted that the decision variable and population FEF activity should initially decrease in all trials but increase again only when the chosen target is in the neuron's response field (Fig. 6m, Supplementary Fig. 9m). However, population FEF activity decreased and then increased in all trials, before it eventually decreased in trials where the animal chose the target away from the neuron's response field, resulting in a small activity bump (Fig. 6n, Supplementary Fig. 9n).

The motor suppression model predicts that information about accumulated evidence should be maintained during the dip, but it also predicts that crossing the decision boundary should not trigger a saccade during the motor suppression interval. Thus, we hypothesized that FEF activity in this model might represent a "motor-decision variable", a version of the decision variable which is scaled-down by a constant factor during the suppression interval. The motor-decision variable encapsulates within a DDM framework the idea that the FEF combines information from accumulated evidence with signals linked to motor output. A saccade is triggered when the motor-decision variable, not the decision variable, crosses the boundary. Unlike the pause or reset models, this can account for the rebound in the FEF activity observed regardless of whether the chosen target is inside or

outside the response field (Fig. 6o, Supplementary Fig. 9o). Therefore, the motor suppression model can parsimoniously account for the FEF population activity as well as the RT dip.

Furthermore, the motor suppression model makes another unique prediction about behavior, namely, that motor suppression triggered by the presample onset might affect other ongoing motor activity, such as microsaccades. As before, the magnitude averaged across trials should be proportional to the probability of detecting a change, and hence, to the coherence. To test this prediction, we examined the microsaccade rate over time for each coherence and presample condition (Fig. 7a, e). We indeed found that for the longest presample duration, there was a coherence-dependent reduction in microsaccade rate, such that high coherence changes elicited a greater reduction in microsaccade rate (Fig. 7d, h). For one monkey, this difference in microsaccade rate was also significant for the 400 ms presample duration (Fig. 7c), and was reduced for the 0 ms presample duration despite an overall increase in microsaccade rate at this time (Fig. 7b). The other monkey had a very low microsaccade rate during the first 600 ms of the task, making it difficult to observe such a reduction (Fig. 7e-g). This dip in microsaccade rate demonstrates that saccadic motor output is inhibited even when it does not involve a response to a target. Thus, four lines of evidence support the motor suppression model: it provides the best fit to the RT distribution, it has a desirable speed-accuracy tradeoff, it uniquely explains both the dip and rebound in FEF activity, and it predicts the dip in microsaccade rate. These converging results of behavior and neural activity provide confidence that motor suppression is an important strategy for changing evidence in perceptual decision-making.

## Discussion

In the present study, we found that changes in evidence strength during a perceptual decision-making task led to a transient suppression of three separate measures, including RT distribution, population and single-neuron FEF activity, and microsaccade rate. Larger changes in evidence elicited a larger dip in all three measures, an observation that goes against classic models in which more evidence always shortens RTs and increases FEF activity for the corresponding behavioral response. By fitting models of three potential cognitive strategies to the behavioral data, we found that transiently suppressing motor output after change detection explains our observations. Critically, it also explains the observations which motivated the other two strategies we examined[13,16,21,33–35]. While the motor suppression mechanism is sufficient to explain our observations, we cannot exclude the possibility of additional mechanisms acting in parallel. Causal experiments which induce a transient suppression of FEF activity could clarify the relationship between evidence integration, motor suppression, and the behavioral and neural dips. However, our results hold regardless of whether the dip in FEF activity causes motor suppression, or whether it simply reflects its presence.

Motor suppression is a general strategy for improving performance in behavioral tasks. Upon observing a change in the environment, briefly suppressing motor activity allows extra time for slower cognitive processes to utilize the new information before committing to a motor action[36]. Changes in the environment are most likely to be observed in our task during trials with high coherences, leading to the observed negative coherence-dependent dip. The mechanisms which drive our model may also explain similar visual phenomena in humans, such as saccade inhibition[17,23,37] and the remote distractor effect[24]. Motor inhibition may be a general mechanism used by humans to deal with surprising stimuli[21,25–27,32,38].

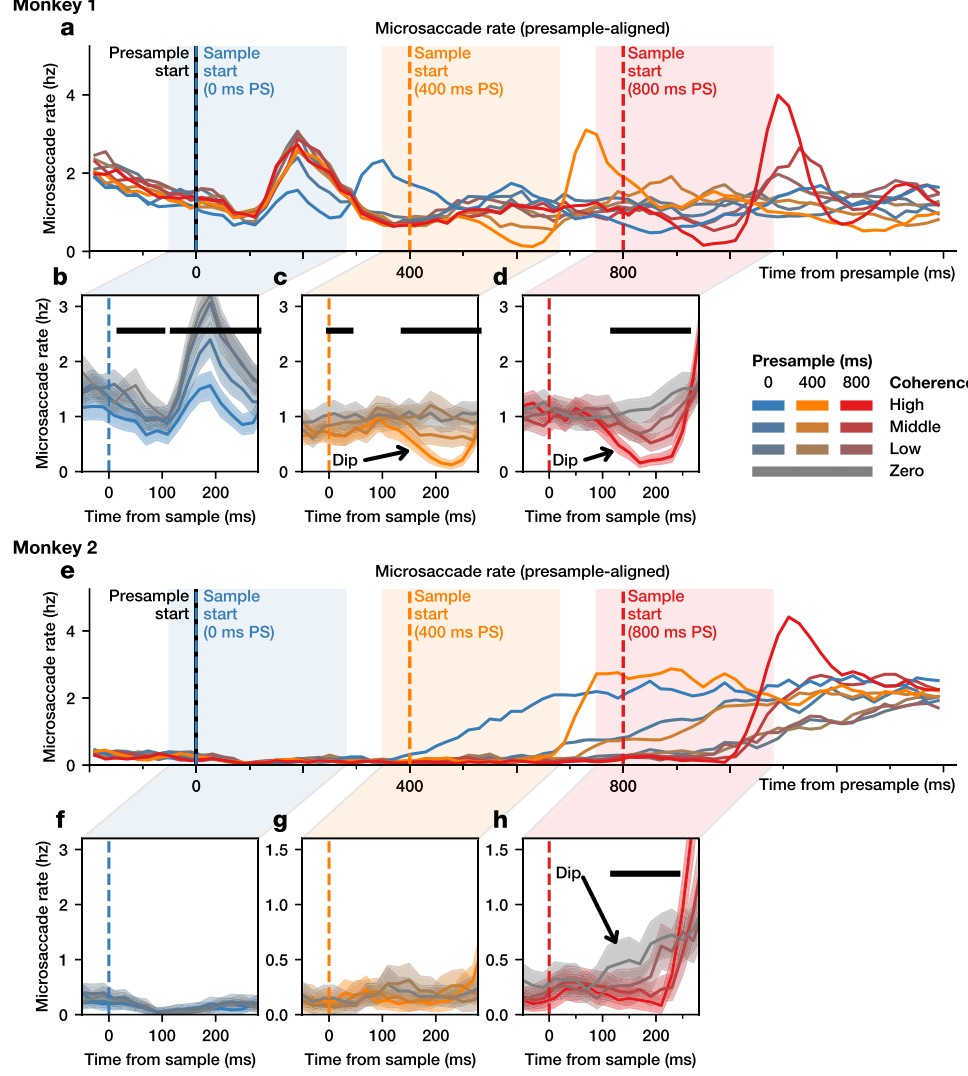

**Fig. 7 Dips in microsaccade rate.** Microsaccade rate is shown for all trials, aligned to the presample onset, for Monkey 1 (**a**) and Monkey 2 (**e**). Highlighted are the microsaccade rates centered around the onset of the sample during trials with 0 ms (**b**, **f**), 400 ms (**c**, **g**), and 800 ms (**d**, **h**) presample for Monkey 1 (**b–d**) and Monkey 2 (**f–h**). The black bar indicates significance ($p < 0.05$, one-tailed test for a decrease in the mean, bootstrapping across neurons). Shaded region represents bootstrapped 95% confidence interval of the mean.

Our results suggest that FEF does not directly encode the decision variable, but rather, it encodes a mixture of the decision variable and a motor signal. This means that FEF might not be directly responsible for integrating evidence, a conclusion supported by causal experiments[39,40] and by the fact that evidence presented during the dip period can still be integrated[13,16,41]. The classic DDM is a one-dimensional model, and we preserved this unidimensionality in our implementation of motor suppression by modeling a motor-decision variable in FEF. While this simplification aids in our analysis, in reality, FEF activity is not one-dimensional, and the mechanisms of motor suppression may show little overlap with those for evidence integration, despite both receiving representation within FEF. For instance, motor suppression appears to mask an independent evidence integration signal. While this work could not address the neurobiological source of motor inhibition, one possibility is that ascending excitatory input synapses onto an inhibitory subpopulation within FEF, providing non-specific lateral inhibition across FEF. The presence of such an inhibitory population within FEF is supported by microstimulation studies, which show that sub-threshold stimulation of regions of FEF outside of the response

field prevents task-related saccades during the stimulation period[42,43], and by experiments showing that cooling FEF leads to increases in the duration of microsaccade suppression[39]. The transient suppression of FEF may be mediated by the release of inhibition[44–47].

Alternatively, motor suppression might arise from reduced ascending input, perhaps mediated by the subthalamic nucleus[48,49], which is common to many regions of the brain. In addition to FEF, a stimulus dip has been observed in LIP[12,50,51], superior colliculus[52,53], visual areas V1 and V2[54], and striatum[55–57]. Similar dips in neural activity have also been observed in FEF during a countermanding task[58] and striatum during an anti-saccade task[56], two tasks which may require motor suppression. In behavioral experiments, the RT dip occurs even after task-irrelevant changes in the visual field[31], after both high contrast changes and isoluminant changes[31], and across the visual field[17,59,60]. These studies collectively suggest that the dip, and motor inhibition more generally, are not localized exclusively to FEF or limited to our specific task, but may be a more general mechanism for dealing with a changing environment. Thus, dips caused by motor suppression may serve more broadly as a marker for attentional shifts or perception of stimulus changes, such

as through the orienting reflex[61–63]. We showed that the neural mechanisms of motor suppression do not significantly overlap with those of motor activity or reward bias. However, the neural mechanisms of motor suppression may not be fully independent of other cognitive processes, so it is possible that there is overlap in the implementation of motor suppression with other processes unrelated to our task. Our results also raise caution about experimental design and interpretation in tasks with changing evidence[5–10]: rather than assuming accumulated or instantaneous evidence correspond directly to brain activity, we must also consider transient effects resulting from the changes in evidence.

Our results suggest that motor suppression is effective because it prevents choices made using outdated evidence. During the task used in our study, the animal's performance was largely at chance level during the first 200 ms after sample onset, in line with the non-decision time in our model, and has comparable timing to the end of the dip in the RT distribution and in FEF activity. Unlike the other models, the motor suppression model slows RTs only in task conditions where it also improves accuracy. We interpret this to mean that motor suppression uses a fast process to halt saccades which were planned using the noisy evidence provided during the presample period, rather than the new higher-coherence evidence. This finding is in line with experiments showing that microstimulation after the onset of saccade planning but before saccade onset slows the resulting saccade[64]. From a normative perspective, this means that the dip mechanism offers the ability to reject planned saccades which were made from incomplete information. By chance, half of these premature saccades will be correct and half will be incorrect. Those that are by chance correct will have a slightly increased RT. Those that would have been by chance incorrect will have an opportunity to be corrected within the motor suppression window. The utility of motor suppression, therefore, is to prevent erroneous responses from being made based on the integrated noise prior to evidence onset. Thus, by interrupting saccade planning for choices based on noise, motor suppression can serve as a general mechanism for increasing accuracy during decision-making in light of new evidence.

## Methods

**Behavioral task.** Two rhesus monkeys, 1 and 2, were trained to perform a two-alternative forced-choice color-discrimination task (Fig. 1) described in Ref. [28]. In each trial of this task, a central square stimulus was presented consisting of a $20 \times 20$ grid of green and blue pixels that rearranged randomly at 20 Hz. The animal indicated its choice by shifting its gaze to one of two flanking choice targets, one green and one blue. The location of one target was chosen based on the response field of the neuron recorded in that session, determined through a memory saccade task, and the other target was opposite to the first. The trial was rewarded via juice delivery if the selected target color corresponded to the majority color of pixels in the sample. Reward cues were displayed surrounding the saccade targets which indicated whether a large or small reward would be delivered for a correct response to the corresponding target. Reward cues were randomly assigned to a target on each trial. 3 (2) drops of juice were given for the large (small) reward condition for Monkey 1, and 3-5 (1-2) drops for Monkey 2.

Stimulus presentation was divided into two consecutive periods containing an uninformative "presample" followed by an informative "sample". There was always an equal number of blue and green pixels displayed during the presample period. Task difficulty was manipulated by parametrically varying the difference in the fraction of pixels of each color in the sample, which we refer to as color coherence. A coherence of 0 indicated equal numbers of both colors, whereas 1 indicated a solid color. No explicit cue was presented to indicate the transition from presample to sample, and the change was instantaneous. The presample duration was selected randomly from three possible time intervals—0, 400 or 800 ms—with equal probability. Animals were allowed to direct their gaze to a choice target any time after the onset of the sample. Eye movements were tracked at 225 Hz with a high speed eye tracker (ET49; Thomas recording). A premature choice before the sample onset aborted the trial, and was punished by a 2-s timeout. In a small fraction of trials (5%), the sample was identical to the presample and maintained zero color coherence throughout the trial. On these trials, animals were allowed to respond at any point during stimulus presentation. The ratio of pixels in the high, medium, and low coherence trials for Monkey 1 were 70:30, 60:40, and 53:47, and

for Monkey 2 were 63:37, 58:42, and 52:48, respectively. We also included a zero-coherence condition for each monkey, for which the ratio was 50:50. The zero-coherence condition therefore did not include a clearly-defined transition from presample to sample. We analyzed 28,378 trials from Monkey 1 and 19,514 trials from Monkey 2. All procedures used in this study were approved by the Institutional Animal Care and Use Committee at Yale University, and conformed to the Public Health Service Policy on Human Care and Use of Laboratory Animals and the Guide for the Care and Use of Laboratory Animals.

**Reaction time distribution and behavioral functions.** The reaction time (RT) distribution and psychometric function were calculated using all the trials in which the monkey successfully completed a saccade to one of the two choice targets. This included the trials with incorrect choices, the trials with choices during the pre-sample period, and the trials in which the animal failed to maintain fixation on the chosen target. The RT histogram was constructed using 20-ms bins and smoothed with an order 1 Savitzky-Golay filter of width 5. We estimated 95% confidence intervals in each coherence condition for visualization by bootstrapping across trials. We performed 10,000 resamplings with replacement from all trials with the given coherence across all sessions, computed the smoothed histogram of each resample, and then found the 2.5% and 97.5% quantiles at each time point.

Statistical significance for the difference in RT histogram at each point in time was determined using a separate bootstrapping procedure which compared the highest and lowest coherence conditions. This consisted of performing 10,000 resamplings with replacement from the highest and the lowest non-zero coherence trials, computing the confidence interval for mean RT difference between the two conditions at each point in time. One-tailed tests were performed to test the null hypothesis that neural activity specifically showed a reduction and not an increase. The lowest non-zero coherence was used instead of zero-coherence trials due to the limited number of zero-coherence trials. The chronometric function was calculated using only correct choices made after the sample onset.

**Analysis of microsaccades.** We detected microsaccades using the method of Ref. [33]. Briefly, the time of microsaccade was determined as the center of an interval in which the eye velocity exceeded six times a robust estimator of the standard deviation. Large saccades, such as the saccades to choice targets and those resulting in fixation breaks, were excluded. The microsaccade rate was calculated in successive 20-ms bins aligned at the presample onset, and then smoothed using an order-1 Savitzky-Golay filter of width 3. One-tailed tests were used to test for a reduction in microsaccades.

**Visualization of population and single-neuron activity.** We recorded from 57 neurons in the frontal eye field (FEF) of Monkey 1 and 23 neurons from FEF in Monkey 2. Neurons were sorted online and tracked throughout the experiment. For each neuron, we constructed a mean spike density function for a given experimental condition by calculating mean spike counts in successive 20-ms bins aligned at the onset of the presample or sample, and smoothing with an order 1 Savitzky-Golay filter of width 3. This was then normalized by subtracting the mean firing rate of each neuron during the presample period of the trials with an 800 ms presample duration for all conditions. These spike density functions were averaged across all neurons together to obtain the population activity. Confidence intervals for visualization of the population activity were estimated by bootstrapping across neurons. We performed 10,000 resamplings with replacement from all neurons across all sessions, computed the smoothed spike counts of each resample, and then found the 2.5% and 97.5% quantiles at each time point. Confidence intervals for statistical significance for the mean difference in spike count between highest and lowest coherence conditions were obtained from a separate bootstrapping procedure, where a sampling distribution of mean spike difference was formed from 10,000 resamplings with replacement of neurons for both conditions. One-tailed tests were used to test for a reduction in neural activity.

Activity shown for individual neurons was computed similarly to the population activity, but smoothed using an order 1 Savitzky-Golay filter of width 5 for visualization, and no normalization was applied.

**Single-neuron regression analysis.** In order to understand how task events and experimental conditions modulated the firing rate of each neuron over time, an ordinary least squares regression model was used to analyze the activity of individual neurons. Spike counts were predicted during three 100-ms intervals: the presample interval ($0 < t < 100$ ms), the sample interval ($P_i + 100 < t < P_i + 200$ ms), and the saccade interval ($S_i - 50 < t < S_i + 50$ ms), where $t$ is the time since presample onset, $P_i$ is the presample duration on trial $i$, and $S_i$ is the time of the saccade on trial $i$. For each interval $I$, we predicted spike counts as

$$x_I^i = \beta_0 + \beta_1 C_i + \beta_2 R_i + \beta_3 F_i \qquad (1)$$

where $C_i$ is the color coherence on trial $i$, $R_i$ is whether the large or small reward target was in the response field ($R_i = 1$ or $-1$, respectively), and $F_i$ is whether or not the choice was into or out of the response field ($F_i = 1$ or $-1$, respectively).

To explore the change in spiking activity in response to task events, we implemented a time-resolved ordinary least squares regression analysis. Spikes were counted using 25-ms bins ($\Delta t = 25$). Spike counts $x_t^i$ at time $t$ (measured from

**Table 1 Constants defining the regression kernels.**

| Kernel ($j$) | Event aligned to | Scaling variable | Kernel start time (ms) ($T^j_{start}$) | Kernel end time (ms) ($T^j_{end}$) |
|---|---|---|---|---|
| P | Presample | | −500 | 2500 |
| PR | Presample | Large reward in response field ($R_i$) | −500 | 2500 |
| E | Sample | | 0 | 300 |
| EC | Sample | Coherence ($C_i$) | 0 | 300 |
| S | Saccade | | −200 | 200 |
| SF | Saccade | Choice in response field ($F_i$) | −200 | 200 |

**Table 2 Constants defining the regression kernels for the modified regression model.**

| Kernel | Event aligned to | Scaling variable | Kernel start time (ms) ($T^j_{start}$) | Kernel end time (ms) ($T^j_{end}$) |
|---|---|---|---|---|
| P* | Presample | | −500 | 2500 |
| PR* | Presample | Large reward in response field ($R_i$) | −500 | 2500 |
| E* | Sample | | 0 | 400 |
| EC* | Sample | Coherence ($C_i$) | 0 | 400 |
| S* | Saccade | | −800 | 200 |
| SI* | Saccade | Choice in response field ($S_i$) | −800 | 200 |
| SC* | Saccade | Coherence | −1000 | 200 |
| SIC* | Saccade | Choice in response field and coherence ($C_i \times S_i$) | −1000 | 200 |

the presample onset) for trial $i$ was predicted as

$$x^i_t = k^P_t + k^{PR}_t R_i + \left( k^E_{t-P_i} + k^{EC}_{t-P_i} C_i \right) \delta_{P_i \neq 0} + k^S_{t-S_i} + k^{SF}_{t-S_i} F_i \quad (2)$$

where, for trial $i$, $P_i$ is the presample duration, $S_i$ is the time of the saccade, $C_i$ is the color coherence, $R_i$ is whether the large or small reward target was in the response field ($R_i = 1$ or $-1$, respectively), $F_i$ is whether or not the choice was into or out of the response field ($F_i = 1$ or $-1$, respectively), and $\delta$ is the indicator function. The $k$ values correspond to kernels aligned to different events; $k^j_0$ is the bin containing the given event for kernel $j$, and $k^j_t$ is the bin $t$ ms relative to the event, where $t \in \left\{ T^j_{start}, T^j_{start} + \Delta t, \dots, T^j_{end} \right\}$ and $k^j_t = 0$ if $t < T^j_{start}$ or $t > T^j_{end}$. Each kernel is identified by a short string, where the first letter indicates the alignment ("P" to presample, "E" to sample or evidence onset, and "S" to saccade) and subsequent letters indicate any conditions ("R" for reward magnitude, "C" for coherence, and "F" for the response field location). Presample-aligned kernels cover the entire trial duration, while sample-aligned kernels and saccade-aligned kernels span a short period surrounding or following the event for sample-aligned and saccade-aligned kernels, respectively. Since here we focus on the effect of evidence change (e.g. transition from presample to sample) on neural modulation, we estimated evidence kernel only for the trials where presample duration was not zero ($\delta_{P_i \neq 0}$). The six kernels in the equation above have constants given in Table 1.

In the main text, we focus our analysis on the coherence-dependent evidence-aligned EC kernel and the presample-aligned P kernel, which are expected to reflect transient reductions, or dips, in FEF activity. Thus, we use the term "evidence kernel" to refer to the EC kernel, and the term "stimulus kernel" to refer to the P kernel minus a baseline, chosen as the mean value of the P kernel in the interval 0-100 ms.

To control for the dip as an artifact of saccadic activity, we fit a separate regression model which differed only in the kernels used (Supplementary Fig. 6a). This model includes extended saccade kernels scaled by the coherence. To distinguish the kernels in this model from those of the previous model, we appended a "*" suffix. The full model is

$$\begin{aligned} x_t = {} & k^{P*}_t + k^{PR*}_t R_i + \left( k^{E*}_{t-P_i} + k^{EC*}_{t-P_i} C_i \right) \delta_{P_i \neq 0} + k^{S*}_{t-S_i} + k^{SC*}_{t-S_i} C_i \\ & + \left( k^{SI*}_{t-S_i} + k^{SIC*}_{t-S_i} C_i \right) S_i \end{aligned} \quad (3)$$

The kernels used for this analysis are shown in Table 2. Likewise, the "evidence* kernel" is defined as the EC* kernel, and the "stimulus* kernel" is defined as the P* kernel minus the mean of the P* kernel in the interval from 0 to 300 ms.

**Analysis of transient activity.** To examine whether the dips in the mean population kernels were representative of kernels at the single neuron level, singular value decomposition (SVD) was performed on the evidence and stimulus kernels across neurons. SVD is a standard technique in linear algebra which is similar in principle to principal component analysis (PCA), but does not normalize the single-neuron kernels by subtracting the mean kernel across the population at each time point. Briefly, for a matrix of kernels $M$, the singular vectors are defined to be the eigenvectors $v_i$ of the matrix $M^T M$, sorted by decreasing eigenvalue. Likewise,

the factor scores for the $i$'th singular vector are defined as the projection of the data onto these singular vectors, namely $Mv_i$. Stimulus kernels were first truncated to the interval from 0 to 300 ms after stimulus onset before performing SVD.

To confirm that the dip in FEF activity was not confounded by pre-saccadic activity, we analyzed the timing of the dip separately for the trials of different RT quintiles. FEF population activity for each RT quintile was computed similarly to the FEF spike density function described previously by using a first-order Savitzky-Golay filter of width 5 (Supplementary Fig. 6d-g, j-l). Fifteen resamplings of the mean spike density function were computed, and the first local minimum for each condition in each resampling was detected by a simple local minimum detection algorithm. The algorithm stepped through the timeseries until encountering a value exceeding the observed minimum value by a fixed tolerance determined empirically to be 0.12 spikes per second. Kendall's tau was used for correlation and statistical tests due to the limited temporal resolution and limited number of quintiles resulting in several ties.

To understand the difference in latency between the dip in the evidence kernel and the dip in the stimulus kernel, we analyzed the mean of each of these kernels. We determined the minimum as the median of the three lowest values in the kernel within 300 ms from the sample or presample onsets for the evidence or stimulus kernels, respectively. We tested whether the minimum of the evidence kernel was later than the stimulus kernel by bootstrapping across neurons. We performed 10,000 resamplings with replacement from all evidence or stimulus kernels, computed the minimum of the mean using the procedure described above, and tested whether the minimum of the stimulus kernel was earlier than that of the evidence kernel.

The reduction in neural activity following sample or presample onset was quantified using the evidence and stimulus dip indices. These were used to provide a direct comparison between the evidence and stimulus dips, and to compare them to other physiological data. Dip indices were computed by z-scoring the kernel and then taking the mean z-scored evidence or stimulus kernel value from an interval $I$. We determined $I$ separately for the evidence and stimulus dip indices by finding the two time points with the largest fraction of significant kernels (Figs. 4d, 7b, Supplementary Figs. 4d and 7b). For the evidence dip, we found $I=[125,175]$ ms for both monkeys. For the stimulus dip index, we found $I=[100,150]$ ms using the same procedure for Monkey 1. The two minimum time points were not consecutive for Monkey 2, so we used the same interval as in Monkey 1.

**Generalized drift-diffusion model.** The reaction times were modeled using the generalized drift-diffusion model[29] (GDDM), which extends the standard drift-diffusion model[1] by allowing the model parameters to be arbitrary functions of time. The form of the GDDM used here is one of the models considered in Ref. [28], and was previously found to have excellent performance during the task used in the present study. The model includes several extensions on the standard DDM to accommodate the temporal and reward structure of the task. For the task's temporal structure, the model includes leaky integration, as well as a delayed linear increase in gain during each trial. For the task's reward structure, the model includes a baseline offset— impacting both starting position and the value to which leaky integration decays—and a "mapping error" in which the high reward choice is sometimes chosen after integrating sufficient evidence for a low reward choice.

The GDDM is given by the equation

$$dx = -l(x + mt)dt + I_{t>D}\Gamma(t)Cs\,dt + \Gamma(t)dW \qquad (4)$$

where $s$ is the signal-to-noise ratio, $l$ is the leak constant, $m$ is the strength of the time-dependent reward bias, $C$ is the coherence, $D$ is the duration of the presample, $W$ is a Wiener process, and $\Gamma(t) = \begin{cases} \gamma_0, & t < t_0 \\ \gamma_0 + m_\gamma(t - t_0), & t \geq t_0 \end{cases}$.

This GDDM model was fit to each monkey separately. It was fit using maximum likelihood on the full distribution through differential evolution[29]. For robustness, fitting was performed using an exponential distribution mixture model with a rate and mixture strength fit to data[28]. The model was simulated by solving the Fokker-Planck equation with a timestep of 5 ms and a space discretization of 0.005.

**Modified GDDM to test potential mechanisms underlying the dip.** In order to understand the relationship between the evidence, stimulus, and RT dips, we constructed modified GDDMs, implementing three potential cognitive mechanisms. The pause model was implemented by setting the drift rate and noise to 0 within an interval, $[t_{start}, t_{stop}]$ ms. The reset model was implemented by setting the leak $l$ to a fixed value $l_{dip}$ within an interval, $[t_{start}, t_{stop}]$ ms, where $l_{dip}$ is fit to the data. The motor suppression model was implemented by introducing a motor-decision variable $x'$ such that, for decision variable $x$,

$$x' = \begin{cases} cx, & t_{start} \leq t \leq t_{stop} \\ x, & \text{otherwise} \end{cases} \qquad (5)$$

where $0 < c < 1$. Here, we arbitrarily set $c = 0.2$. We assume that, while the decision variable $x$ continues to track integrated evidence, we only trigger a decision when the motor-decision variable $x'$ crosses the boundary. The motor-decision variable combines two separate processes – evidence integration and motor suppression – into a single decision variable to produce predicted FEF activity. In the neural implementation of motor suppression, it is likely that the integration process and the motor suppression process would evolve independently of each other, and are only combined into the motor decision variable in the final stage. For the purpose of efficient simulation, we use an equivalent formulation whereby the bound is increased by a factor of $1/c$ during the interval from $[t_{start}, t_{stop}]$ ms. In order to mitigate numerical artifacts caused by the abrupt change in bound, the change in bound height was implemented to be a smooth increase and decrease in the form of the probability density function of the Beta(3,3) distribution.

We assume that these dips were more likely to occur in trials with a higher color coherence, with a probability determined by a saturating sigmoidal curve with a fixed scale which was fit to the data. The probability of detecting the change on any given trial (and therefore invoking the given dip mechanism) was

$$p_{detect} = 2/(1 + \exp(-\lambda C)) - 1 \qquad (6)$$

where $C$ is the color coherence ranging from 0 for an equal pixel ratio to 1 for a solid color. A total of three additional parameters were fit for the pause and motor suppression models, and four additional parameters for the reset model. Four GDDMs were constructed, one for each of the three dip mechanisms described above, and one baseline containing none of the dip mechanisms. Parameters were fit for each model by simulating with the Crank-Nicolson (pause and reset) or implicit (motor suppression) method, using differential evolution to optimize the likelihood over the full probability distribution[29]. Measurements of model BIC were computed using this full distribution likelihood. To compute ΔBIC, we subtracted each modified GDDM's BIC from the BIC of the unmodified GDDM.

To gain insights into the potential function of the dips in perceptual decision-making, we evaluated the impact of each dip mechanism on the RT and accuracy by comparing the RT and accuracy of each of the modified GDDMs to those from the same modified GDDM but with the modification disabled, i.e. keeping the same parameters for all other aspects of the model.

Predicted FEF activity was determined by transforming the decision variable trace. For each model, we simulated the monkey's highest coherence when the correct response was inside or outside the response field, with an 800-ms presample and the large reward target inside the response field. We recorded the time evolution of the decision variable distribution, and took the mean at each point in time to obtain a deterministic prediction of the decision variable value. For the motor suppression model, while we simulated by increasing the integration bounds, we interpret this equivalently as a temporary multiplicative suppression of the decision variable by the corresponding amount. As a result, FEF activity was predicted by dividing the decision variable value by the height of the bound. Finally, to simulate the spike density functions, decision variable traces were thresholded at 0 and filtered with a Gaussian kernel. Predicted RT distributions were likewise determined by computing the first passage time of the GDDM and smoothing with a Gaussian kernel.

Simulations were performed using the PyDDM package[29] on Python 3. Stimulus presentation was controlled by custom software designed for our experimental setup.

**Reporting summary**. Further information on research design is available in the Nature Research Reporting Summary linked to this article.

## Data availability

The GDDMs from this study, including online demos, are available in the PyDDM documentation at: https://pyddm.readthedocs.io/en/latest/cookbook/papers/dip.html. Other datasets generated during and/or analyzed during the current study are available from the corresponding author by request. Source data are provided with this paper.

## Code availability

All code used to generate the figures in this paper is available at https://github.com/mwshinn/figures_from_dip_paper.

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

## Acknowledgements

We thank Matthew Gay, Joey Schnurr and Irina Bobeica for their technical support. This study was supported by the National Institute of Health (R01 MH108629; R01 MH112746). MS was supported by the Gruber Foundation Science Fellowship.

## Author contributions

D.L. and H.S. designed and performed the experiments. M.S., J.D.M., and H.S. analyzed the data. All authors interpreted the results. M.S. and D.L. wrote the paper. All authors reviewed, edited, and approved the manuscript.

## Competing interests

The authors declare no competing interests.
