## [Peer Review File · Nature Communications]

Transient neuronal suppression for exploitation of new sensory evidenceREVIEWER COMMENTS

Reviewer #1 (Remarks to the Author):

This manuscript provides a thorough investigation of how non-stationary sensory evidence impacts saccade behavior and neural responses in the frontal eye field (FEF) during decision-making. The authors report the occurrence of an evidence-dependent suppression of saccadic RTs and of FEF activity following the presentation non-stationary visual stimuli. Computational analyses suggest that this dip in FEF activity plays a motor suppression role, downregulating motor output transiently. Importantly, the finding that micro-saccades (which are irrelevant to the task) are also suppressed in an evidence-dependent way following stimulus presentation supports the latter interpretation. The paper is well written and easy to follow despite the complexity of the analyses. I hope the comments below will help strengthening the manuscript.

Major:

1. The manuscript would touch a broader readership and gain in impact if it were integrated in a broader literature than in its current state. The authors should also highlight further the novelty of the study with respect to past research.

For instance, the introduction states "in most previous neurophysiological studies, stimulus evidence had constant magnitude within a single trial and evidence onset was clearly demarcated." However, a number of recent neurophysiological and neuroimaging studies have exploited dynamic decision-making tasks, in which evidence is not constant in magnitude but is rather evolving over time during deliberation. Here are some examples: Thura and Cisek, 2014 and 2017, *Neuron*; Thura et al., 2014; Khalighinejad et al., 2020, *PNAS* and *Neuron*; Derosiere et al., 2019, *J Neurophysiol*; Van Maanen et al., 2016, *NeuroImage*; Wyart et al., 2012, *Neuron*; Pares-Puljors et al., 2021, *NeuroImage*. I agree with the authors that these studies did not investigate how evidence alters transient neural responses to imperative stimuli (except maybe Pares-Puljors et al., 2021, *NeuroImage*, investigating the impact of evidence on transient EEG potentials in humans). Hence, I do not question the novelty of the study, but I believe that it may be beneficial for the article to contrast the present study with the former work.

The introduction also cites non-primate studies reporting the occurrence of dips in FEF and LIP activity following stimulus occurrence. Could the authors broaden the scope of this literature to other species? Is there any evidence for such a suppression in humans? Wessel and Aron (2017, *Neuron*) provides a review of how unexpected perceptual events may affect neural activity in humans. Most noticeably, a transient suppression of corticospinal excitability has been reported in the same time window as the one investigated by the authors (150 ms post-stimulus; see Wessel and Aron, 2013, *J Neurosci*). On a related note, I think that it would be possible to highlight what is novel in the present study. Two novel findings are the coherence-dependent effect and the modelling outcomes. The former finding is not well introduced and could leave the reader with the impression that the results presented from figure 2 to 5 are just different ways of showing the presence of a dip, which has already been shown in the past, while, in fact, these figures show how the evidence-dependent effect is robust across the dataset.

2. I also wonder if it would be possible to discuss a bit more about the potential functional role of such an evidence-dependent motor suppression mechanism. I think the discussion would gain some explanatory power if this issue was addressed. During decision-making under conflict (or in the face of ambiguous sensory evidence), suppression is thought to allow one to take time to choose the right action (see Wessel and Aron, 2016). Yet, here a stronger suppression is observed when evidence is the highest, which could be counterintuitive, and is thus quite interesting.

3. Relatedly, do the authors have a possible explanation for why early (micro) saccades are mostly observed and suppressed in the 800 ms PS condition? Is it possible that these responses reflect timed guesses of the monkeys (which are only fully possible in the 800 ms condition, because here the evidence onset can be anticipated)? If the authors agree with this, it may explain why FEF suppression

after evidence onset is stronger with increasing coherence and potentially with higher reward: Monkeys may have a latent tendency to perform early guesses after evidence onset (especially after a period of waiting in the 800 ms condition), because premature responses before evidence onset are time-punished. Monkeys might suppress these "uninformed saccades" however if a change in evidence strength suggests high reward probability for "informed saccades" (in case of highly coherent evidence). In this case, a general trend for increasing FEF activity over time (Figure 3) might be reflective of increased "urgency" to respond with little or no evidence, which is suppressed (FEF activity dip) if changes in evidence strength suggest high reward probability for informed responses. To test whether these early saccades after evidence onset really form timed guesses, one may look at the choice accuracy as a function of RT in the 800 ms SOA condition. Another way to possibly test this is look at the histogram of premature responses in the interval of -400 to 0 ms before evidence onset: One would expect a disproportional number of premature responses shortly before evidence onset, if these form timed guesses. Is the winning GDDM model reconcilable with this interpretation on monkey's behavior?

4. Can the winning GDDM model replicate RT histograms in Figure 3 and capture differences in frequency of early saccades between experimental conditions? Demonstrating this congruence between predicted and actual behavior will substantiate the modelling results. It seems that the comparison between actual and predicted data is restricted to the FEF activity in the condition of interest (i.e. large reward, 800 ms PS and high coherence trials, Figure 6k), but it may be necessary to demonstrate that the model generally captures the behavior of the monkeys well during this task.

5. When introducing the different models, could authors explain how the suggested mechanisms (pause, reset, motor suppression) may lead to improvement in choice performance? We see that this is the case (Figures 6e-f), but to me it is not clear how "pause" and "motor suppression" may reduce noise in the evidence accumulation process. Related to this, all models seem to improve behavior only in small reward trials, why is this the case? Do authors agree that the "reset" model (Monkey 1) and the "pause" model (Monkey 2) lead to strongest improvements in choice accuracy as the evidence signal becomes more coherent?

6. The RT histogram was constructed using 20-ms bins and smoothed with an order 1 Savitzky-Golay filter of width 5. Could it be possible that the absence of dip in the RT distribution for the low evidence conditions was related to the size of the bins and/or the smoothing procedure? Do the authors get the same findings using different bin sizes and smoothing? To be mentioned in the supplementary section?

7. All figures in the main text focus on Monkey 1 data. It would be informative if some of the figures could show data from Monkey 2 as well. Monkey 2 data are a bit less convincing, hence their appearance in supplementary materials, but a reader who do not consult the supplementary materials may get a biased vision of the findings, missing the between-subject variability in the effects reported.

Minor:

1. In the abstract and introduction, the FEF area is mentioned before we know that the study is concerned with saccade decisions. It might be useful to first mention that the study is about saccade decisions before mentioning the FEF.

2. How many trials were analyzed for each monkey?

3. The use of one-tailed t-tests for some of the analyses should be motivated.

4. On the findings presented in lines 247-253 (related to figure S6): I am not sure that a correlation is the most appropriate analysis here given the small number of points. If I understood well, the correlation was performed on the 5 RT bins. I do not question the validity of the conclusions of this section, but the authors should maybe consider using another statistical test, comparing directly the dip minimum time between each RT bin.

5. Lines 345-349: it may be unclear why the pause model does not predict a dip in FEF activity.

6. Line 351: "Chosen target" instead of "chosen choice"?

7. What does the zero coherence condition reflect in the figures (especially for the 800 ms condition)?

8. Figure 6d: it may be useful to mention briefly in the main text and/or figure legend why a delta BIC is presented in the figure (and why the higher the delta BIC, the better the model; the latter information could be added in the Methods section). Also, it may be a bit unclear why the pause and the motor suppression models show similar delta BIC while their performance in predicting FEF responses and behavior is different. Could the authors elaborate quickly on this in the results section?

9. In Figure 7b-d, there seems to be an increase in micro saccade rate from 100 to 300 ms in the 0 ms PS condition. For the same time interval of 100-300 ms, micro saccades seem to be suppressed for 400 ms PS and 800 ms PS conditions. Any idea why this might be the case?

Julie Duque, Gerard Derosiere & Thomas Carsten

Reviewer #2 (Remarks to the Author):

The manuscript by Shinn et al. identifies a phenomenon in behavior and neural activity associated with decision making. They find that the onset of evidence accumulation is associated with a decrease in neural activity ("dip") as well as a retardation of fast behavioral responses. The paper notes that this phenomenon is not consistent with basic predictions of drift-diffusion models of decision making that predict increases in neural activity and speeding up of behavioral responses.

Understanding the computational logic of the dip is important as this is a widespread phenomenon in neurophysiological experiments of decision making. Moreover, the experimental setting is suitable for tackling this question since the task nicely dissociates stimulus onset time from the time relevant evidence has to be accumulated.

I have no major concern about the analyses or the results. However, I would like to encourage the authors to think harder about what their finding means. As is, the paper provides an account of the dip phenomenon but does not explain why the brain might adopt this mechanism. I think the paper would be stronger if it considers two lines of thinking to address this shortcoming.

1. Can there be a normative account for this particular solution? For example, is there some cost function (e.g., robustness, accuracy, etc) that this solution minimizes?

2. Can this be an epiphenomenon that results from the constraints of a recurrent system handling the necessary computations? For example, are there constraints that force a recurrent neural network to exhibit this phenomenon?

I think this latter point is particularly important to further assess. The analyses offered in the paper evaluates the dip in terms of modulations of a (motor) decision variable. I am not sure this interpretation is correct. I think it is likely that the population responses shortly after evidence onset are structured differently from the later response associated with evidence accumulation. If so, interpreting the dip on the same footing as the later response can be misleading.

To address this point, the authors should analyze the covariance matrix of signals across the population of recorded neurons (no need to be simultaneous as this can be done on PSTHs), and assess whether the covariance matrix is similar early on and later after in the evidence accumulation

period. Alternatively, they can assess how well the principal components derived from early responses capture variance in late responses and vice versa. If the responses early and late are structured similarly, then the authors' interpretations seem valid. If not, the authors should adjust the language they use and consider the dip as a process distinct from evidence accumulation (that possibly masks the underlying evidence accumulation). Such an observation would indicate that the dip is the consequence of the system being pushed by the stimulus in a direction away from the movement initiation state.

Finally, I think it would be good to evaluate this result in the context of early work by Churchland and Shenoy related to retardation of motor response after perturbations, which is interpreted in terms of the system moving away from a movement onset state.

It was a pleasure to read your manuscript!

Reviewer #1:

This manuscript provides a thorough investigation of how non-stationary sensory evidence impacts saccade behavior and neural responses in the frontal eye field (FEF) during decision-making. The authors report the occurrence of an evidence-dependent suppression of saccadic RTs and of FEF activity following the presentation non-stationary visual stimuli. Computational analyses suggest that this dip in FEF activity plays a motor suppression role, downregulating motor output transiently. Importantly, the finding that micro-saccades (which are irrelevant to the task) are also suppressed in an evidence-dependent way following stimulus presentation supports the latter interpretation. The paper is well written and easy to follow despite the complexity of the analyses. I hope the comments below will help strengthening the manuscript.

Major:

1. The manuscript would touch a broader readership and gain in impact if it were integrated in a broader literature than in its current state. The authors should also highlight further the novelty of the study with respect to past research.

We greatly appreciate the reviewers' suggestions on how to expand the reach of the paper so that it appeals to a larger audience. Changes related to each individual suggestion raised by reviewers are shown below.

For instance, the introduction states "in most previous neurophysiological studies, stimulus evidence had constant magnitude within a single trial and evidence onset was clearly demarcated." However, a number of recent neurophysiological and neuroimaging studies have exploited dynamic decision-making tasks, in which evidence is not constant in magnitude but is rather evolving over time during deliberation. Here are some examples: Thura and Cisek, 2014 and 2017, *Neuron*; Thura et al., 2014; Khalighinejad et al., 2020, *PNAS* and *Neuron*; Derosiere et al., 2019, *J Neurophysiol*; Van Maanen et al., 2016, *NeuroImage*; Wyart et al., 2012, *Neuron*; Pares-Puljors et al., 2021, *NeuroImage*. I agree with the authors that these studies did not investigate how evidence alters transient neural responses to imperative stimuli (except maybe Pares-Puljors et al., 2021, *NeuroImage*, investigating the impact of evidence on transient EEG potentials in humans). Hence, I do not question the novelty of the study, but I believe that it may be beneficial for the article to contrast the present study with the former work.

We agree with the reviewer that there should be a discussion about the relationship between our task and other tasks with changing evidence, even if they do not directly examine transient effects during evidence integration.

We added the following to the Introduction:

While several studies have utilized time-varying stimuli (Wyart et al 2012; Thura and Cisek 2014; Glaze et al 2015; van Maanen et al 2016; Derosiere et al 2019; Khalighinejad et al 2020), it is unknown how moment to moment changes in sensory inputs impact evidence integration. In the present study, we investigated how the trajectory of neural activity related to evidence accumulation might be adjusted by the subtle onset of sensory signals decoupled from the onset of the stimulus itself.

We also added the following to the Discussion:

Our results also raise caution about experimental design and interpretation in tasks with changing evidence (Wyart et al 2012; Thura and Cisek 2014; Glaze et al 2015; van Maanen et al 2016; Derosiere et al 2019; Khalighinejad et al 2020): rather than assuming accumulated or instantaneous evidence correspond directly to brain activity, we must also consider transient effects resulting from the changes in evidence.

The introduction also cites non-primate studies reporting the occurrence of dips in FEF and LIP activity following stimulus occurrence. Could the authors broaden the scope of this literature to other species? Is there any evidence for such a suppression in humans? Wessel and Aron (2017, Neuron) provides a review of how unexpected perceptual events may affect neural activity in humans. Most noticeably, a transient suppression of corticospinal excitability has been reported in the same time window as the one investigated by the authors (150 ms post-stimulus; see Wessel and Aron, 2013, J Neurosci).

We agree with the reviewer that there is a compelling case for motor suppression in humans, and our discussions should not be limited to the NHP literature.

We added the following to the Introduction:

Finally, decision makers might adapt to the unpredicted arrival of new information simply by suppressing their motor outputs (Purcell et al. 2010; Bompas and Sumner 2011; Bompas et al. 2015) without modifying the state of decision variables, similar to models proposed in other systems in humans (Reingold and Stampe 2002; Wessel et al 2013; Wessel et al 2017; Duque et al 2017).

We added the following to the Discussion:

The mechanisms which drive our model may also explain similar visual phenomena in humans, such as saccade inhibition (Reingold and Stampe 2002; Bompas and Sumner 2011) and the remote distractor effect (Bompas and Sumner 2015). Motor inhibition may be a general mechanism used by humans to deal with surprising stimuli (Corbetta and Shulman 2002; Wessel and Aron 2013; Wessel and Aron 2017; Duque et al 2017; Salinas and Stanford 2018; Salinas et al 2019).

On a related note, I think that it would be possible to highlight what is novel in the present study. Two novel findings are the coherence-dependent effect and the modelling outcomes. The former finding is not well introduced and could leave the reader with the impression that the results presented from figure 2 to 5 are just different ways of showing the presence of a dip, which has already been shown in the past, while, in fact, these figures show how the evidence-dependent effect is robust across the dataset.

We thank the reviewer for drawing our attention to the fact that the coherence-dependent effect is given less spotlight in the introduction than the modeling work. It is especially important to get this right because the modeling has a longer history than the negative coherence-dependence effect.

In the Abstract, we add that the dip is “*sensitive to stimulus strength*” to emphasize the coherence-dependent effect. (The abstract already mentions the modeling results.)

In the Introduction, we have added the following passage to emphasize the novelty of the coherence-dependent effect.

After the onset of the informative stimulus, we observed transient suppression in saccadic motor output, as well as in FEF activity at a population and single-neuron level. Moreover, suppression of motor output and FEF activity was greater for stronger sensory signals, resulting in a negative relationship of the RT distribution and FEF activity on coherence. This is in contrast to the usual positive relationship of coherence with these measurements.

2. I also wonder if it would be possible to discuss a bit more about the potential functional role of such an evidence-dependent motor suppression mechanism. I think the discussion would gain some explanatory power if this issue was addressed. During decision-making under conflict (or in the face of ambiguous sensory evidence), suppression is thought to allow one to take time to choose the right action (see Wessel and Aron, 2016). Yet, here a stronger suppression is observed when evidence is the highest, which could be counterintuitive, and is thus quite interesting.

We thank the reviewer for drawing our attention to this point. We agree that suppression allows one to take the time to make the right action, and see how this could be seen to conflict with the observation that stronger suppression occurs after the highest coherence stimuli. We address this in two ways. First, we clarify in the main text the fact that our mechanisms are probabilistic, an important detail we should not have buried in the Methods. Second, we emphasize this interpretation that suppression allows taking the time to choose the right action.

First, to clarify that our mechanisms are probabilistic, we first add the following to the Results section when the models are first introduced.

We also assume that the ability to detect a change in evidence is probabilistic, with a probability of detection increasing linearly with coherence.

When describing the analysis of the neural data in the Results section, we clarified our justification of choosing the high coherence trials in terms of the probability of observing the dip, bringing attention to the fact that we interpret detection of a change as probabilistic:

For this analysis, we focused on the trials expected to show the strongest dip and the highest probability of detecting the change in evidence, namely, those with 800-ms presample duration, high coherence, and the large reward target in the neuron's response field (Figure S10).

Also, when describing the analysis of microsaccades, we explicitly mention the probability of detecting a change in relation to our previous results:

As before, the magnitude averaged across trials should be proportional to the probability of detecting a change, and hence, to the coherence.

Second, to bring attention to the relationship between motor activity allowing extra time to make the decision, and the probabilistic nature of our mechanisms, we add the following to the Discussion:

Motor suppression is a general strategy for improving performance in behavioral tasks. Upon observing a change in the environment, briefly suppressing motor activity allows extra time for slower cognitive processes to utilize the new information before committing to a motor action (Wessel et al 2016). Changes in the environment are most likely to be observed in our task during trials with high coherences, leading to the observed negative coherence-dependent dip.

We also emphasize this interpretation of motor suppression when it is described in the Results section:

This mechanism ensures slow cognitive processes have sufficient time to integrate new information before making a choice.

3. Relatedly, do the authors have a possible explanation for why early (micro) saccades are mostly observed and suppressed in the 800 ms PS condition? Is it possible that these responses reflect timed guesses of the monkeys (which are only fully possible in the 800 ms condition, because here the evidence onset can be anticipated)?

Yes, we agree with the reviewer's interpretation that there appears to be something like timed guess: the monkeys know not to guess early on, because a later display of evidence may make the choice obvious. However, as time passes, they make the best guess they can given the information they have acquired, consistent with an urgency signal.

We added the following to the Results section:

...trials with shorter presample durations contained few responses within a 200-ms window after sample onset (Figure 2a,c), presumably related to fewer responses driven by a slowly ramping urgency signal (Thura et al 2012; Shinn et al 2020)

We also add a supplemental figure showing that the conditions with the highest baseline activity are those that show the dip most strongly.

Figure S10: Dip magnitude is largest at highest activity. The dip is shown for all presample and coherence conditions for trials where the large or small reward target is inside and outside the response field (RF).

If the authors agree with this, it may explain why FEF suppression after evidence onset is stronger with increasing coherence and potentially with higher reward: Monkeys may have a latent tendency to perform early guesses after evidence onset (especially after a period of waiting in the 800 ms condition), because premature responses before evidence onset are time-punished. Monkeys might suppress these "uninformed saccades" however if a change in evidence strength suggests high reward probability for "informed saccades" (in case of highly coherent evidence). In this case, a general trend for increasing FEF activity over time (Figure 3) might be reflective of increased "urgency" to respond with little or no evidence, which is suppressed (FEF activity dip) if changes in evidence strength suggest high reward probability for informed responses. To test whether these early saccades after evidence onset really form timed guesses, one may look at the choice accuracy as a function of RT in the 800 ms SOA condition. Another way to possibly test this is look at the histogram of premature responses in the interval of -400 to 0 ms before evidence onset: One would expect a disproportional number of premature responses shortly before evidence onset, if these form timed guesses. Is the winning GDDM model reconcilable with this interpretation on monkey's behavior?

With respect to saccades, we agree with the reviewer's interpretation that motor suppression works to suppress "uninformed saccades", which are most prominent surrounding evidence onset. Since the dip occurs after evidence onset, suppressed responses would occur shortly after evidence onset. We show, as suggested by the reviewer, that responses immediately after evidence onset primarily consist of timed guesses with approximately chance accuracy, and the accuracy only begins to increase about 200

ms after evidence onset. This means that motor suppression is primarily suppressing uninformed saccades.

We add a panel c to Figure S1 showing the psychometric function extended over time. We can see that responses within the first 200 ms after evidence onset are approximately chance, followed by a sharp increase in accuracy.

Figure S1: Behavioral data and model prediction. (a-b) Psychometric (a) and chronometric (b) functions. Markers indicate data, and lines indicate the predictions of the motor suppression GDDM. 95% confidence interval error bars are hidden beneath the markers. (c) Psychometric function extended over time in 50 ms bins. Time points are hidden for bins without at least one correct and one incorrect response. Error bars represent 95% confidence intervals computed with a normal approximation.

We also point this out in the Results section as follows:

This strategy is effective because it reduces the fraction of responses occurring during the first 200 ms, which are close to chance (Figure S1c).

To emphasize the increase in overall guesses, and the fact that this can be gleaned by examining the zero-coherence RT distribution, we include the following in the Results section:

All three models showed the strongest improvement in accuracy for 800 ms presample trials, for which the animal is making the most random guesses, as indicated by the elevated responses in the zero-coherence condition (Figure 2a,c).

With respect to microsaccades, we believe, as hinted by the reviewer, that there can be no reduction in microsaccades without some baseline level of microsaccades. Monkey 2 makes few microsaccades during the first 600 ms of the task, so there is little room to show effects of motor suppression.

We add the following to the Results section:

The other monkey had a very low microsaccade rate during the first 600 ms of the task, making it difficult to observe such a reduction (Figure 7e-g).

4. Can the winning GDDM model replicate RT histograms in Figure 3 and capture differences in frequency of early saccades between experimental conditions? Demonstrating this congruence between predicted and actual behavior will substantiate the modelling results. It seems that the comparison between actual and predicted data is restricted to the FEF activity in the condition of interest (i.e. large reward, 800 ms PS and high coherence trials, Figure 6k [now Figure 6o]), but it may be necessary to demonstrate that the model generally captures the behavior of the monkeys well during this task.

We thank the reviewer for this important suggestion to improve the strength of our argument. We see two important ways in which the model may capture the behavioral data. First, it may do so at the large-scale level by capturing overall trends in accuracy and response time. Second, critically, it may reproduce the coherence-dependent dip. We incorporate into the manuscript evidence that the motor suppression model is able to replicate both large-scale and transient behavioral phenomena.

First, we describe the large-scale fit of the data through the use of the psychometric and chronometric functions. In Figure S1a-b (above), we show the monkey's psychometric and chronometric functions and compare them to those of the motor suppression model. We update the figure legend accordingly to say the following:

Markers indicate data, and lines indicate the predictions of the motor suppression GDDM.

We also updated the Results section as follows:

Additionally, the motor suppression model is able to capture the psychometric and chronometric functions (Figure S1a,b).

Second, we demonstrate that all three models considered here reproduce the coherence-dependent dip phenomenon in the RT distribution. We extend Figure 6 by adding the new panels e-h as follows:

Figure 6: Comparison of computational models of the dip. (a-c) Schematic of the (a) pause model, (b) reset model, and (c) motor suppression model across stages of the decision-making process. (d) The fit of each GDDM to the RT distribution, as quantified by BIC, is shown for each of the three models. (e-g) The simulated RT distribution for 800 ms presample at the time of evidence onset is shown for the (e) pause, (f) reset, and (g) motor suppression models. (h) The RT distribution for Monkey 1 in 800 ms presample trials at the time of evidence onset. (i-k) For each coherence, presample, and reward condition, the difference in mean RT and accuracy, with and without the dip mechanism, are plotted for the (i) pause, (j) reset, and (k) motor suppression models. (l-n) Neural predictions of the (l) pause model, (m) reset model, and (n) motor suppression model, based on the mean decision variable or motor-decision variable (see Methods) for inside (in-RF, black) and outside (out-RF, red) the response field on correct high-coherence trials for models fit to the RT distribution of Monkey 1. (o) Population

activity from FEF neurons in Monkey 1 for correct responses inside (gray) and outside (red) the response field for high-coherence, 800-ms presample trials.

We refer to this figure in the text using the following:

Each of the models considered was able to fit the coherence-dependent behavioral dip in the RT distribution (Figure 6e-h).

5. When introducing the different models, could authors explain how the suggested mechanisms (pause, reset, motor suppression) may lead to improvement in choice performance? We see that this is the case (Figures 6e-f [now Figure 6i-j]), but to me it is not clear how "pause" and "motor suppression" may reduce noise in the evidence accumulation process.

We thank the reviewers for this suggestion, and agree that an intuitive explanation for each model will improve the strength of the paper.

We have changed the description of each model in the Results section to include a sentence on why the model makes sense intuitively, as follows:

First, in the “pause model” (Figure 6a), when a change is detected, the stream of evidence is briefly interrupted, or “paused”. This mechanism can be thought of as blocking volatile information in order to focus on a more stable evidence signal. Second, in the “reset model” (Figure 6b), the change elicits a partial “reset” of the decision variable back towards its initial value. This mechanism effectively discards any noise which may have been integrated during the presample period. Finally, in the “motor suppression model” (Figure 6c), motor output is temporarily blocked, thereby suppressing responses during this period without impacting the decision variable. This mechanism ensures slow cognitive processes have sufficient time to integrate new information before making a choice.

We also elaborated on the effectiveness of the motor suppression model in the discussion section as follows:

Our results suggest that motor suppression is effective because it prevents choices made using outdated evidence. During the task used in our study, the animal’s performance was largely at chance level during the first 200 ms after sample onset, in line with the non-decision time in our model, and has comparable timing to the end of the dip in the RT distribution and in FEF activity. Unlike the other models, the motor suppression model slows RTs only in task conditions where it also improves accuracy. We interpret this to mean that motor suppression uses a fast process to halt saccades which were planned using the noisy evidence provided during the presample period, rather than the new higher-coherence evidence. This finding is in line with experiments showing that microstimulation after the onset of saccade planning but before saccade onset slows the resulting saccade (Churchland and Shenoy 2007). From a normative perspective, this means that the dip mechanism offers the ability to reject planned saccades which were made from incomplete information. By chance, half of these premature saccades will be correct and half will be incorrect. Those that are by chance correct will have a slightly increased RT. Those that would have been by chance incorrect will have an opportunity to be corrected within the motor suppression window. The utility of motor suppression, therefore, is to prevent erroneous responses from being made based on the integrated noise prior to evidence onset. Thus, by interrupting saccade planning for choices

based on noise, motor suppression can serve as a general mechanism for increasing accuracy during decision-making in light of new evidence.

Related to this, all models seem to improve behavior only in small reward trials, why is this the case?

The accuracy is already very high for the high reward trials, and therefore, there is less room for improvement. Likewise, accuracy is lowest for 800 ms presample trials, and hence, these also show the largest improvement with motor suppression for both high and low reward trials.

We updated the Results section as follows:

All three models showed the strongest improvement in accuracy for 800 ms presample trials, for which the animal is making the most random guesses, as indicated by the elevated responses in the zero-coherence condition (Figure 2a,c).

Do authors agree that the "reset" model (Monkey 1) and the "pause" model (Monkey 2) lead to strongest improvements in choice accuracy as the evidence signal becomes more coherent?

While these three mechanisms may lead to the largest improvement in choice accuracy, we realize that the speed-accuracy tradeoff figure does not capture our intended message. As reviewers mention, the reset model for Monkey 1 and the pause model for Monkey 2 show the strongest improvement in mean choice accuracy for our task. However, the reason for showing the speed-accuracy tradeoff in Figure 6i-k was twofold. First, we wanted to show that all three models did improve accuracy at the expense of RT. Second, more subtly, only the motor suppression model improved accuracy for each condition that it slowed. In other words, while all three models increased the mean accuracy and produced longer RTs, the pause and reset models had several individual conditions whereby the RT was longer but there was no improvement in accuracy. Thus, in these two models, the speed-accuracy tradeoff occurred at the level of the task, rather than the level of individual conditions. By contrast, the motor suppression model resulted in longer RTs only for the conditions in which it also improved accuracy. If motor suppression is a general mechanism for responding to surprising information, then it should not impose a global penalty on all decision-making, only on decisions for which it is able to improve the accuracy.

We rewrote the section of the Results discussing Figure 6i-k to be the following:

Next, we show that the motor suppression model exhibits a speed-accuracy tradeoff for all conditions in the task. All three models showed a speed-accuracy tradeoff when considering mean performance in the task as a whole, resulting in increased mean accuracy but slower mean RT (Figure 6i-k, S9i-k, black cross). However, a general mechanism for decision-making should ideally not slow RT in environments where it does not improve accuracy. We examined this by considering each task condition separately. All three models showed the strongest improvement in accuracy for 800 ms presample trials, for which the animal is making the most random guesses, as indicated by the elevated responses in the zero-coherence condition (Figure 2a,c). In addition, the motor suppression model resulted in slower RT only for conditions where it also improved accuracy (Figure 6k, S9k). By contrast, the pause and reset models slowed RT in some conditions without improving their accuracy (Figure 6i,j, S9i,j). This shows how the motor suppression model can serve as an efficient general mechanism for increasing accuracy without creating a generalized deficit in speed.

6. The RT histogram was constructed using 20-ms bins and smoothed with an order 1 Savitzky-Golay filter of width 5. Could it be possible that the absence of dip in the RT distribution for the low evidence conditions was related to the size of the bins and/or the smoothing procedure? Do the authors get the same findings using different bin sizes and smoothing? To be mentioned in the supplementary section?

We agree with the reviewer that some readers may raise doubt about the impact of smoothing on the results. Since all statistics were computed on the unsmoothed RT distribution (and FEF activity), the smoothing does not impact the statistical significance. While the bin size may impact statistical significance, we chose 20 ms as an appropriate tradeoff between granularity and statistical power.

First, we emphasize in the main text that statistical tests were performed on the unsmoothed histograms, by adding the following to the figure legend:

RT distributions are smoothed for visualization only, with significance tests performed before smoothing.

Second, we add the following supplemental figure:

Figure S2: RT dip is robust across bin sizes and smoothing widths. For each monkey, we constructed the RT histogram immediately after sample onset for 800 ms presample trials, analogous to Figure 2b,d, with varying bin sizes and smoothing widths, inset in each figure. Smoothing width was defined as the filter width of the order 1 Savitzky-Golay filter. “no filter” indicates the absence of smoothing. The box indicates the bin size and filter width chosen for the present study.

We refer to this figure the text as follows:

The dip was present for a wide range of bin sizes (Figure S2).

7. All figures in the main text focus on Monkey 1 data. It would be informative if some of the figures could show data from Monkey 2 as well. Monkey 2 data are a bit less convincing, hence their appearance in supplementary materials, but a reader who do not consult the supplementary materials may get a biased vision of the findings, missing the between-subject variability in the effects reported.

We agree with the reviewer that it would be useful for readers to see the range of individual variation between monkeys. The reason Monkey 2's data appeared in the supplement is not because it is less convincing, but rather, to maintain clarity for the reader.

We have extended Figures 2 and 3 to describe Monkey 2's RT and FEF dips, and extended Figure 7 to include Monkey 2's microsaccade data. These are shown below:

Figure 2: Transient effect of changes in evidence on RT. (a,c) The RT distribution for all trials, aligned to the presample onset, for Monkey 1 (a) and Monkey 2 (c). (b,d) The RT distribution centered around the onset of the sample during trials with 800-ms presample for Monkey 1 (b) and Monkey 2 (d). (e) Predictions from a generalized drift-diffusion model (GDDM) for the portion of the RT distribution shown in (b,d). The black bar indicates significance ($p < 0.05$, one-tailed test for a decrease in the mean, bootstrapping across trials). Shaded regions represent bootstrapped 95% confidence interval. RT distributions are smoothed for visualization only, with significance tests performed before smoothing.

Figure 3: Transient effect of changes in evidence on population FEF activity. (a,e) The normalized population activity for all trials, aligned to the presample onset, for Monkey 1 (a) and Monkey 2 (e). (b-d,f-h) Highlighted is activity centered around the onset of the sample during trials with a 0- (b,f), 400- (c,g), and 800-ms (d,h) presample duration for Monkey 1 (b-d) and Monkey 2 (f-h). The black bar indicates significance ($p < 0.05$, one-tailed test for a

decrease in the mean, bootstrapping across neurons). Shaded regions represent bootstrapped 95% confidence interval. (i-k) Predictions from a drift-diffusion model decision variable (DV) are shown below for the 0- (i), 400- (j) and 800-ms (k) presample durations. Light gray indicates no prediction within the model's non-decision time. Activity is smoothed for visualization only, with significance testing performed before smoothing.

Figure 7: Dips in microsaccade rate. (a) Microsaccade rate is shown for all trials, aligned to the presample onset, for Monkey 1. Highlighted are the microsaccade rates centered around the onset of the sample during trials with (b) 0 ms, (c) 400 ms, and (d) 800 ms presample. The

black bar indicates significance ($p < 0.05$, one-tailed test for a decrease in the mean, bootstrapping across neurons). Shaded region represents bootstrapped 95% confidence interval.

Minor:

1. In the abstract and introduction, the FEF area is mentioned before we know that the study is concerned with saccade decisions. It might be useful to first mention that the study is about saccade decisions before mentioning the FEF.

We changed the abstract to say the following:

Here, we trained monkeys to identify and respond via saccade to the dominant color of a dynamically refreshed bicolor patch...

2. How many trials were analyzed for each monkey?

We added the following to the methods:

We analyzed 28,378 trials from Monkey 1 and 19,514 trials from Monkey 2.

3. The use of one-tailed t-tests for some of the analyses should be motivated.

We changed the descriptions in the figure captions in the Results section to the following:

one-tailed test for a decrease in the mean

We added the following to the Methods section for the RT distributions:

One-tailed tests were performed to test the null hypothesis that neural activity specifically showed a reduction and not an increase.

We added the following to the Methods section for microsaccades:

One-tailed tests were used to test for a reduction in microsaccades.

We added the following to the Methods section for neural activity:

One-tailed tests were used to test for a reduction in neural activity.

4. On the findings presented in lines 247-253 (related to figure S6): I am not sure that a correlation is the most appropriate analysis here given the small number of points. If I understood well, the correlation was performed on the 5 RT bins. I do not question the validity of the conclusions of this section, but the authors should maybe consider using another statistical test, comparing directly the dip minimum time between each RT bin.

We agree with the reviewer that a Spearman correlation significance test performed on only five points is not the most robust way of testing for a trend.

We instead perform the Kendall tau correlation on the trajectories of all resampled iterations of the 5 RT bins. Kendall tau is able to handle ties, and thus we are able to utilize all of the data instead of just the median. We also provided more detail on our method for producing correlations and p-values in the figure legend. The figure and its legend are as follows:

Figure S6: The evidence dip is not saccade-related. (a) A schematic of the alternative regression model, with kernels designated by a * suffix. (b,c,h,i) Format similar to Figure 4d,

except using the evidence kernel instead of the evidence kernel, for Monkey 1 (b,c) and 2 (h,i). (d,j) Normalized FEF activity from high-coherence 800 ms presample trials is aligned to the sample onset and plotted separately for each of 5 RT bins for Monkey 1 (d) and 2 (j). (e,k) Normalized FEF activity from these trials is aligned to the saccade onset and plotted separately for each of 5 RT bins for Monkey 1 (e) and 2 (k). (f,m) Time of the first local minimum after the sample onset for the curves in (e) and (k) are plotted for Monkey 1 (f) and 2 (m) against the mean RT of the trials in the RT bin. Points indicate the median of 15 resamplings of the FEF activity, and error bars indicate the interquartile range. Inset is the Kendall's tau correlation r and the corresponding p -value across all resamplings. (g) Time of the first local minimum before the saccade onset for the curves in (f) are plotted for Monkey 1 against the mean RT of the trials in the RT bin. Inset is the Kendall's tau correlation r and the corresponding p -value across all resamplings. Minima could not be reliably detected in for saccade-aligned activity in Monkey 2.*

We describe this in the Methods with the following:

Kendall's tau was used for correlation and statistical tests due to the limited temporal resolution and limited number of quantiles resulting in several ties.

5. Lines 345-349: it may be unclear why the pause model does not predict a dip in FEF activity.

We added the following to the Results:

The pause model assumes that incoming evidence is briefly disregarded, and that the decision variable should remain fixed at its present value. Thus, it predicts a flattening of the decision variable trace and hence a flattening of FEF activity, followed by an increase for trials with the chosen target inside the response field and a decrease for those with the chosen target outside the response field (Figure 6l, S9l).

6. Line 351: "Chosen target" instead of "chosen choice"?

We changed line 351 as suggested.

7. What does the zero coherence condition reflect in the figures (especially for the 800 ms condition)?

We added the following to the description of the task in the Results:

For comparison, we also included a "zero-coherence" condition, in which an equal ratio of blue and green pixels was maintained throughout the entire trial.

We also added the following to the Methods:

We also included a zero-coherence condition for each monkey, for which the ratio was 50:50. The zero-coherence condition therefore did not include a clearly-defined transition from presample to sample.

8. Figure 6d: it may be useful to mention briefly in the main text and/or figure legend why a delta BIC is presented in the figure (and why the higher the delta BIC, the better the model; the latter information could be added in the Methods section). Also, it may be a bit unclear why the pause and the motor

suppression models show similar delta BIC while their performance in predicting FEF responses and behavior is different. Could the authors elaborate quickly on this in the results section?

We expanded our description of the results in Figure 6d by adding the following paragraph to the Results:

First, we show the motor suppression model provided the best fit to the RT distribution. Each of the models considered was able to fit the coherence-dependent behavioral dip in the RT distribution (Figure 6e-h). Therefore, we evaluated the fit of each model to the data using Bayesian information criterion (BIC). For comparison, we evaluated the BIC of the model without any dip mechanism present. The improvement in the fit of the model to the monkey's behavior can be quantified by the difference in BIC between the model with and without the dip mechanism (Δ BIC). We found that the improvement in fit is greatest for the motor suppression model (Figure 6d, S9d). Additionally, the motor suppression model is able to capture the psychometric and chronometric functions (Figure S1a,b). Therefore, the motor suppression model is best able to explain the dip present in the RT distribution.

Additionally, we provided the following summary at the end of the modeling section to draw attention to the fact that the ability to fit the RT distribution and the fit to the neural data are two distinct lines of evidence supporting the model:

Thus, four lines of evidence support the motor suppression model: it provides the best fit to the RT distribution, it has a desirable speed-accuracy tradeoff, it uniquely explains both the dip and rebound in FEF activity, and it predicts the dip in microsaccade rate.

9. In Figure 7b-d, there seems to be an increase in micro saccade rate from 100 to 300 ms in the 0 ms PS condition. For the same time interval of 100-300 ms, micro saccades seem to be suppressed for 400 ms PS and 800 ms PS conditions. Any idea why this might be the case?

The reviewer points out a fascinating observation: even though there is an overall increase in microsaccade rate in Monkey 1 for 0 ms presample trials, this reduction is smaller for high coherence trials than it is for low coherence trials. Despite the fact that this is quantitatively consistent with other presample durations, i.e., the largest coherence is associated with the lowest microsaccade rate, it differs qualitatively, as it is associated with an increase in microsaccade rate instead of a decrease. It appears that the coherence-dependent decrease in microsaccade rate associated with the dip is compounded with a coherence-independent increase in microsaccade rate associated with some unknown process related to trial onset. We do not have sufficient data to speculate on the unknown process resulting in this increase, and on why it occurs only in one monkey, but perhaps it is related to attentional effects.

We added the following to the Results section to explicitly highlight this observation:

For one monkey, this difference in microsaccade rate was also significant for the 400 ms presample duration (Figure 5c), and was reduced for the 0 ms presample duration despite an overall increase in microsaccade rate at this time (Figure 5b).

Julie Duque, Gerard Derosiere & Thomas Carsten

Thank you all for your helpful comments!

Reviewer #2:

The manuscript by Shinn et al. identifies a phenomenon in behavior and neural activity associated with decision making. They find that the onset of evidence accumulation is associated with a decrease in neural activity ("dip") as well as a retardation of fast behavioral responses. The paper notes that this phenomenon is not consistent with basic predictions of drift-diffusion models of decision making that predict increases in neural activity and speeding up of behavioral responses.

Understanding the computational logic of the dip is important as this is a widespread phenomenon in neurophysiological experiments of decision making. Moreover, the experimental setting is suitable for tackling this question since the task nicely dissociates stimulus onset time from the time relevant evidence has to be accumulated.

I have no major concern about the analyses or the results.

We thank the reviewer for the kind words regarding our manuscript.

However, I would like to encourage the authors to think harder about what their finding means. As is, the paper provides an account of the dip phenomenon but does not explain why the brain might adopt this mechanism.

We agree that the paper would benefit from a more detailed explanation of why the brain would adopt this mechanism. The benefit of motor suppression, as well as the pause and reset models, could be explained more thoroughly.

In addition to the reviewer's specific suggestions below, we have extended the paragraph in the Results introducing the models to be the following:

First, in the "pause model" (Figure 6a), when a change is detected, the stream of evidence is briefly interrupted, or "paused". This mechanism can be thought of as blocking volatile information in order to focus on a more stable evidence signal. Second, in the "reset model" (Figure 6b), the change elicits a partial "reset" of the decision variable back towards its initial value. This mechanism effectively discards any noise which may have been integrated during the presample period. Finally, in the "motor suppression model" (Figure 6c), motor output is temporarily blocked, thereby suppressing responses during this period without impacting the decision variable. This mechanism ensures slow cognitive processes have sufficient time to integrate new information before making a choice.

I think the paper would be stronger if it considers two lines of thinking to address this shortcoming.

1. Can there be a normative account for this particular solution? For example, is there some cost function (e.g., robustness, accuracy, etc) that this solution minimizes?

We agree with the reviewer that a normative account would strengthen the paper. However, providing such an account is difficult not only practically, but also conceptually. We make two primary claims

about the motor suppression model: first, that it improves the mean accuracy in our task, potentially at the expense of a longer RT, and second, that it does not lengthen RT for a particular decision unless it increases accuracy. The former is not unique to the motor suppression model, and is thus unlikely to be useful for normative purposes. Since the latter is more difficult to quantify mathematically, we take a slightly different approach to achieve a normative perspective.

The reason motor suppression does not increase RT is because it operates in the time between stimulus onset and motor response, what would be considered the non-decision time in a DDM. It works to interrupt saccades which were planned before more evidence was made available. From a normative perspective, this means that the dip mechanism offers the ability to reject planned saccades which were made from incomplete information. By chance, half of these saccades will be correct and half will be incorrect. Those that are by chance correct will have a slightly increased RT. Those that would have been by chance incorrect will have an opportunity to be corrected within the motor suppression window.

We added the following paragraph to the Discussion to explain this perspective:

Our results suggest that motor suppression is effective because it prevents choices made using outdated evidence. The monkey performs at approximately chance level during the first 200 ms after sample onset, in line with the non-decision time in our model, and has comparable timing to the end of the dip in the RT distribution and in FEF activity. Unlike the other models, motor suppression only slows RTs in task conditions where it also improves accuracy. We interpret this to mean that motor suppression uses a fast process to halt saccades which were planned using the noisy evidence provided during the presample period, rather than the new higher-coherence evidence. This finding is in line with experiments showing that microstimulation after the onset of saccade planning but before saccade onset slows the resulting saccade (Churchland and Shenoy 2007). From a normative perspective, this means that the dip mechanism offers the ability to reject planned saccades which were made from incomplete information. By chance, half of these saccades will be correct and half will be incorrect. Those that are by chance correct will have a slightly increased RT. Those that would have been by chance incorrect will have an opportunity to be corrected within the motor suppression window. The utility of motor suppression, therefore, is to prevent erroneous responses from being made based on the integrated noise prior to evidence onset. Thus, by interrupting saccade planning for choices based on noise, motor suppression is able to serve as a general mechanism for increasing accuracy during decision-making in light of new evidence.

2. Can this be an epiphenomenon that results from the constraints of a recurrent system handling the necessary computations? For example, are there constraints that force a recurrent neural network to exhibit this phenomenon?

I think this latter point is particularly important to further assess. The analyses offered in the paper evaluates the dip in terms of modulations of a (motor) decision variable. I am not sure this interpretation is correct. I think it is likely that the population responses shortly after evidence onset are structured differently from the later response associated with evidence accumulation. If so, interpreting the dip on the same footing as the later response can be misleading.

To address this point, the authors should analyze the covariance matrix of signals across the population of recorded neurons (no need to be simultaneous as this can be done on PSTHs), and assess whether the covariance matrix is similar early on and later after in the evidence accumulation period. Alternatively, they can assess how well the principal components derived from early responses capture variance in late responses and vice versa. If the responses early and late are structured similarly, then the authors' interpretations seem valid. If not, the authors should adjust the language they use and consider the dip as a process distinct from evidence accumulation (that possibly masks the underlying evidence accumulation). Such an observation would indicate that the dip is the consequence of the system being pushed by the stimulus in a direction away from the movement initiation state.

The reviewer brings up several related but distinct points in this section. We respond to each of these below.

(a) First, the reviewer suggests that the dip may be an epiphenomenon. By definition, an epiphenomenon is “a secondary phenomenon accompanying another and caused by it”. The suggestion that the dip is an epiphenomenon, therefore, can have multiple interpretations.

One interpretation is that the neural dip exhibited in our paper is indicative of, but not causally related to, motor suppression. This question is interesting, but not accessible by the methods presented here. Answering this question would need a causal manipulation of the neural activity, which is beyond the scope of this study.

We have clarified this by adding the following to the Discussion section:

Causal experiments which induce a transient suppression of FEF activity could clarify the relationship between evidence integration, motor suppression, and the behavioral and neural dips.

A second interpretation of the dip as an epiphenomenon is that the neural and behavioral dips exhibited in our paper are not useful for distinguishing different cognitive mechanisms for changing evidence in perceptual decision-making. We believe this is false: as shown in Figures 6 and 7, a model which explains the behavioral and neural dips is also able to explain a rebound in activity for FEF neurons outside the receptive field, provide the best fit to behavioral data, and predict the distribution of microsaccades after changes in evidence.

To emphasize that the dip is indicative of the animal's motor suppression strategy regardless of the neural implementation of evidence integration, we have added the following to the Results:

These converging results of behavior and neural activity provide confidence that motor suppression is an important strategy for changing evidence in perceptual decision-making.

We also add the following to the Discussion:

However, our results hold regardless of whether the dip in FEF activity causes motor suppression, or whether it simply reflects its presence.

A third interpretation of the dip as an epiphenomenon is that any strategy which efficiently solves our task, when implemented within a recurrent system, must exhibit a dip. The reviewer specifically mentions an implementation in an RNN. While a complete characterization of this question is beyond

the scope of this paper (and this response letter), we found this suggestion intriguing and trained a recurrent neural network on our task. If the dip is an epiphenomenon of *any* recurrent system which is capable of solving our task, then such a dip should occur in the trained RNN. However, the trained RNN does not exhibit the behavioral or neural dip. We present full details and methods in Response Supplement 1.

Lastly, we would like to clarify that, even if it were the case that the behavioral and/or neural dips were epiphenomena caused by the rearrangement of population activity, our analyses here demonstrate that it is functionally associated with the suppression of motor output. Therefore, calling the dip an “epiphenomenon” or “not an epiphenomenon” may be a false dichotomy.

(b) The reviewer suggests that language used in the manuscript should clearly indicate that motor suppression is distinct from evidence integration. We fully agree that these are separate processes within the brain.

We have added the following to the Discussion section to make this more clear.

The classic DDM is a one-dimensional model, and we preserved this unidimensionality in our implementation of motor suppression by modeling a motor-decision variable in FEF. While this simplification aids in our analysis, in reality, FEF activity is not one-dimensional, and the mechanisms of motor suppression may show little overlap with those for evidence integration, despite both receiving representation within FEF. For instance, motor suppression appears to mask an independent evidence integration signal.

We add the following to the Results section when the motor-decision variable is introduced to underscore that this is a simplification of reality:

The motor-decision variable encapsulates within a DDM framework the idea that the FEF combines information from accumulated evidence with signals linked to motor output.

Finally, we clarify the scope of the interpretability of the motor-decision variable within the Methods:

The motor-decision variable combines two separate processes – evidence integration and motor suppression – into a single decision variable to produce predicted FEF activity. In the neural implementation of motor suppression, it is likely that the integration process and the motor suppression process would evolve independently of each other, and are only combined into the motor decision variable in the final stage.

(c) The reviewer also noted that it is unlikely that the neural population is structured identically during the dip as it is during evidence integration. In other words, the neural subspace which contains the dip does not coincide with that which drives evidence integration.

We agree with the reviewer, and apologize that this was not made clear within the manuscript. In fact, our data show a clear separation of the dip from evidence accumulation. In Figure 6o, the dip occurs both for trials when the choice is inside the receptive field (resulting in a large positive increase in activity) and outside the receptive field (resulting in a large decrease in activity). If these occurred in the same unidimensional subspace, the dip in one of these two cases would need to be replaced by an “inverted dip”, or a brief elevation in activity.

To emphasize this multidimensionality, we show an example of two simple dimensions with different patterns of activity: the sum of activity between choices inside the receptive field and choices outside the response field, and the difference between them (Figure R1). While the former shows the dip as well as activity during the presample, the latter shows no dynamical changes until after about 200 ms after sample onset.

Figure R1: Two simple axes of FEF activity. The sum (left) and difference (right) of FEF activity for choices inside and outside the response field. Color hues indicate presample duration, and color saturation indicates coherence.

We believe our data can shed some light on how different types of signals are represented and combined in FEF. Several distinct signals are known to be represented within FEF, including those related to motor activity, spatial location, reward, value, choice, accumulated evidence, salience, and many others. Thus, we can generalize the reviewer’s question to the following: to what extent does the dip subspace overlap with the subspaces encoding these other signals within FEF?

We do not believe that this information can be gleaned by analyzing the covariance matrices at different timepoints throughout the trial, because the population structure before and after the dip should be expected to appear very different due to confounding eye movements. After the change in evidence, the number of saccades greatly increases. FEF activity is known to contain distinct and overlapping populations of cells in the FEF which respond to visual stimuli and eye movement. Therefore, the motor responses confound the interpretation of a covariance matrix of (pseudo)population activity before and after the change in evidence. Furthermore, these analyses are confounded by the fact that the dip in neural activity is easiest to measure in cells with a high firing rate, because they have a higher dynamic range to exhibit a reduction in activity. In summary, it is hard to draw robust and rigorous conclusions about the neural implementation of evidence integration and motor suppression from the present data. Such questions are beyond the scope of this study, and may be more accessible to studies involving high-density electrophysiology with simultaneous neuronal recordings, allowing trial-by-trial analyses.

Instead, we focus on two signals which can be reliably extracted from our dataset: reward encoding and motor preparatory signals. We assessed the overlap of the dip subspace with reward and motor signals

by assessing the alignment between the dip's axis and the axes which best explain choice and reward. We quantified this for each cell using the regression coefficients from Figure S3—for choice, we used the choice coefficient at saccade onset, and for reward, we used the reward coefficient at presample onset. Because neuronal recordings were not performed simultaneously, we reasoned that an alignment in the axes could be quantified by a cell-wise Spearman correlation between each neuron's weight on these measures. We found no significant correlation between the strength of the dip and the coding for choice (Spearman $r=-0.11$, $p=0.34$) or reward (Spearman $r=-0.20$, $p=0.07$) (Figure R2). Therefore, we conclude that there is minimal, if any, alignment between the dip axis and these two axes.

Figure R2: For each neuron, the dip index is plotted against the neuron's choice selectivity (left) and the neuron's reward selectivity (right). Both monkeys are shown together.

Overall, these results indicate a minimal alignment of the dip axis with other task-related axes, presenting a complex picture of the neural implementation of motor suppression and perceptual decision-making. Our DDM is a one-dimensional model, but the brain's implementation cannot be one-dimensional. This emphasizes the importance of distinguishing the strategies discussed in our manuscript from their implementation within the brain.

We add a summary of these results to the Results section:

Third, we examined whether the pattern of neural activity associated with the dip is correlated across neurons with the pattern associated with other task elements, such as motor response or reward bias. We computed the Spearman correlation between the non-time-resolved regression coefficients from Figure S4. If these patterns covary across FEF neurons, we would expect a significant positive or negative correlation between the coefficient for coherence at sample onset and either the coefficient for choice at saccade onset or reward at presample onset. However, there was no significant correlation with motor (Spearman $r=-0.11$, $p=0.34$) or reward (Spearman $r=-0.20$, $p=0.07$).

To draw further attention to the interpretation of the motor decision variable and the neural implementation occurring in a complex neural state space, we add the following to the Discussion:

We showed that the neural mechanisms of motor suppression do not significantly overlap with those of motor activity or reward bias. However, the neural mechanisms of motor suppression may not be fully independent of other cognitive processes, so it is possible that there is overlap in the implementation of motor suppression with other processes unrelated to our task.

Finally, I think it would be good to evaluate this result in the context of early work by Churchland and Shenoy related to retardation of motor response after perturbations, which is interpreted in terms of the system moving away from a movement onset state.

We thank the reviewer for the suggestion to discuss the Churchland/Shenoy work.

We have added a discussion of this paper, and preparatory activity activity in general, to the paragraph described in response to Point 1.

It was a pleasure to read your manuscript!

-Mehrdad Jazayeri

Thank you for your insightful review!

Response Supplement 1: RNN methods and results

Results:

The dips in RT distribution and FEF activity do not appear to be general properties of systems which are able to perform our task. In theory, it is possible that the dips emerge as epiphenomena caused by the structure of our task. To determine whether this is the case, we train a recurrent neural network (RNN) to perform our task. Coherence-dependent dip-like signals in the RNN after evidence onset would indicate the dip is an artifact of network rearrangement in response to a strong transient stimulus. We constructed a RNN trained on an adapted version of our task (Figure S3a), and found it captured qualitative properties of the psychometric and chronometric functions (Figure RS1b,c). However, we found no evidence of a coherence-dependent dip in the response time distribution (Figure RS1d), in the mean motor output (Figure RS1e), or in the mean activity of the hidden recurrent units of the network (Figure RS1f). Thus, the dip in RT distribution and in FEF activity are unlikely to be epiphenomena caused by the task structure.

Figure RSI: The dip does not occur in recurrent neural network (RNN) models. (a) Schematic describing the RNN model. Three example inputs to the network are shown on the left, and their corresponding outputs are shown on the right. (b-c) The trained RNN was tested, and the psychometric (b) and chronometric (c) functions are shown. (d) Response time (RT) distribution of the trained network at the time of sample onset for 20 step presample (left) and 40 step presample (right). (e) Output of the motor signal, before discretizing into RT, of the trained network at the time of sample onset for 20 step presample (left) and 40 step presample (right). Responses are shown in the positive and negative directions for coherences of magnitudes 0, 0.5, 1, and 4. (f) Mean hidden unit activity of the trained network at the time of sample onset for 20 step presample (left) and 40 step presample (right). Responses are shown in the positive and negative directions for coherences of magnitudes 0, 0.5, 1, and 4.

Methods:

A recurrent neural network (RNN) was trained on an adapted version of our task. Networks were constructed using the PsychRNN package (Ehrlich et al 2021; eNeuro). We assumed a single stream of Gaussian noise with unit variance as evidence input, and one “motor” choice output. The input stream began on each trial with a mean of zero, corresponding to the presample period. After the duration of the presample period, which could be 0, 20, or 40 timesteps, the sample period involved a shift of the mean evidence input into the positive or negative direction by an amount equal to the coherence, which here was 0.25, 0.5, 1, 2, 4, 6, 8, and 10. The task adapted for RNN did not include a reward bias. The network was trained to output a 0 during the presample period, corresponding to central fixation, and a

1 or -1 after the onset of the sample depending on the mean sign of evidence, corresponding to an eye movement to a target. Training utilized the loss function of Zhang et al (2020; ECAI), which has statistical properties encouraging the network to make a choice towards -1 or 1 and fixate on that value. The choice was considered to be made when the absolute value of the “motor” output signal exceeded a fixed threshold of 0.8. The sign at the time of threshold crossing was considered to be the choice, and the time at which threshold was crossed was the RT. While this means the task was not strictly a forced choice task, in practice the network made a choice over 98% of the time, comparable to the frequency of lapse trials in monkeys. The network contained 20 hidden units with ReLU activation functions. The motor output neuron likewise utilized a ReLU activation function. The network was trained in 50,000 steps with batch size 128, and 64,000 trials were simulated from the trained network. Since step sizes are discrete, the RT distribution was not binned, but it was smoothed with a Savitzky-Golay filter of width 3.

REVIEWERS' COMMENTS:

Reviewer #1 (Remarks to the Author):

The authors have addressed all our comments. We have no further suggestion.

Reviewer #2 (Remarks to the Author):

The authors have thoroughly addressed the issues raised by the reviewers. I think the manuscript is ready for publication.

I only wish to clarify my point about "epiphenomenon" (apology for lack of clarity). The point I wanted to draw the authors' attention to was the possibility that the dip (and the consequent motor suppression) may have no normative account (no "intended" functional role), and instead, it may be an "unintended" consequence of how the stimulus interacts with the state of the system during the wait time. The modeling with a RNN model does not reject or validate this possibility. But it may be possible to address this experimentally. For example, one can test if the relationship between FEF dip and the stimulus would hold if the animal is asked to respond with a manual reach. According to the proposed interpretation, the same suppression should now only be present in the neural circuits that control reaching and not FEF.

Reviewer 1

The authors have addressed all our comments. We have no further suggestion.

We thank the reviewer for their favorable assessment of our manuscript.

Reviewer 2

The authors have thoroughly addressed the issues raised by the reviewers. I think the manuscript is ready for publication.

I only wish to clarify my point about “epiphenomenon” (apology for lack of clarity). The point I wanted to draw the authors’ attention to was the possibility that the dip (and the consequent motor suppression) may have no normative account (no “intended” functional role), and instead, it may be an “unintended” consequence of how the stimulus interacts with the state of the system during the wait time. The modeling with a RNN model does not reject or validate this possibility. But it may be possible to address this experimentally. For example, one can test if the relationship between FEF dip and the stimulus would hold if the animal is asked to respond with a manual reach. According to the proposed interpretation, the same suppression should now only be present in the neural circuits that control reaching and not FEF.

We thank the reviewer for the clarification on what was meant by "epiphenomenon". We agree with this interpretation, and also agree that further experiments, such as the one proposed by the reviewer, could test these predictions explicitly.